# Highly hydrated paramagnetic amorphous calcium carbonate nanoclusters as an MRI contrast agent

Liang Dong[1,2,3,9], Yun-Jun Xu[4,9], Cong Sui[1,9], Yang Zhao[1], Li-Bo Mao ®[1], Denis Gebauer ®[5], Rose Rosenberg[6], Jonathan Avaro[7], Ya-Dong Wu[3], Huai-Ling Gao ®[1], Zhao Pan[1,2], Hui-Qin Wen[8], Xu Yan[3], Fei Li[3], Yang Lu ®[3] ✉, Helmut Cölfen ®[6] ✉ & Shu-Hong Yu ®[1] ✉

Amorphous calcium carbonate plays a key role as transient precursor in the early stages of biogenic calcium carbonate formation in nature. However, due to its instability in aqueous solution, there is still rare success to utilize amorphous calcium carbonate in biomedicine. Here, we report the mutual effect between paramagnetic gadolinium ions and amorphous calcium carbonate, resulting in ultrafine paramagnetic amorphous carbonate nanoclusters in the presence of both gadolinium occluded highly hydrated carbonate-like environment and poly(acrylic acid). Gadolinium is confirmed to enhance the water content in amorphous calcium carbonate, and the high water content of amorphous carbonate nanoclusters contributes to the much enhanced magnetic resonance imaging contrast efficiency compared with commercially available gadolinium-based contrast agents. Furthermore, the enhanced $T_1$ weighted magnetic resonance imaging performance and biocompatibility of amorphous carbonate nanoclusters are further evaluated in various animals including rat, rabbit and beagle dog, in combination with promising safety in vivo. Overall, exceptionally facile mass-productive amorphous carbonate nanoclusters exhibit superb imaging performance and impressive stability, which provides a promising strategy to design magnetic resonance contrast agent.

Amorphous calcium carbonate (ACC), which exists widely in nature, plays a key role as transient precursor in the early stages of biomineral formation[1–4]. Using a bioinspired strategy, diverse materials with controlled phase, physical and chemical properties can be achieved[5,6]. The highly hydrated content is a distinctive feature of ACC, and plays a pivotal role in its stabilization[6–12]. As a metastable phase of calcium carbonate, ACC is unstable in aqueous solution and will transform into crystalline phases rapidly owing to dehydration, ion binding and other factors[6–8,13]. Therefore, the potential application of highly hydrated ACC is largely ignored, and there are few successful examples to utilize ACC in biomedicine.

One significant parameter to enhance the contrasting performance of gadolinium-based $T_1$ MRI contrast agents is their hydration[14]. With seven non-paired electrons, the gadolinium ion possesses a large magnetic moment and long electron spin relaxation time, leading to numerous clinically available extracellular gadolinium-based contrast agents for $T_1$ MRI[14,15]. In virtue of versatile functionalization to interact with biomolecules in vivo and lower gadolinium ion leakage rates benefitting from an inorganic nanostructure, Gd-based inorganic nanoagents attracted considerable attention[15,16]. Unfortunately, the hydration of gadolinium-based nanoparticles suffers from high-temperature synthesis, though the consequent ion leakage is

minimized by the confined nanostructure compared with chelate complexes[15].

Herein, we introduce gadolinium ions into the ACC mineralization process, which have been proven to be integrated into the final amorphous calcium carbonate phase. In this amorphous system, gadolinium ions in conjunction with poly(acrylic acid) facilitate enhanced hydration water and stability of nanoclusters, while the confinement of gadolinium ions by carbonate improve the bio-compatibility and performance, indicating the resulting product has noteworthy MRI contrast agent properties. Furthermore, a mutual effect between the paramagnetic lanthanide gadolinium ion and amorphous calcium carbonate is discovered, which contributes to the maximized hydrated content in the as-prepared amorphous composite nanoclusters and a high longitudinal relaxation. The final para-magnetic amorphous carbonate nanoclusters (ACNC) possess a high water to Ca ratio (water/Ca = 7.2) compared with the normal ACCs (ratios remained constant at about 0.4–1.9). The longitudinal relaxivity of ACNC ($37.93 \pm 0.63$ mM$^{-1} \cdot$ s$^{-1}$ under 3.0 T) has also benefited from the high water content, which is ten times higher than that of the commercially available MR contrast agent gadopentetic acid (Gd-DTPA) and highly resistant to ion leakage so it can serve as a potential MR contrast agent.

## Results

### Characterization of ACNC

So far, besides different kinds of organic additives such as biomole-cules and polymers, magnesium is the only inorganic cation that has been reported capable of stabilizing ACC[6,17]. The gadolinium ion, whose ionic radius is closer to that of the calcium ion than that of the magnesium, can be regarded as a smaller calcium ion analog with higher valence and hydration energy[18,19]. Similarly to the chelation between poly(acrylic acid) (PAA) and calcium for the enhanced stabi-lity of ACC[7,20], carboxylate groups in PAA have the potential to chelate gadolinium ions, allowing the bound complex to later be solidified into carbonate[21].

As shown in Fig. 1a, calcium chloride and gadolinium chloride were mixed with PAA, and equimolar sodium carbonate solution was then added under robust stirring. This efficient synthesis at room temperature proved to yield ACNC in the presence of both gadolinium and PAA. Two liters of aqueous product can be facilely produced, indicating the scalability and reproducibility of the method reported here (Fig. 1b). The presence of clusters dispersed in normal saline was reflected by the Tyndall effect (Fig. 1c). As shown in Fig. 1d, the sphe-rical morphology was further investigated by cryotransmission elec-tron microscopy (cryo-TEM), and proved to be comparable with ACC as reported previously[22]. The nanoclusters observed by STEM (Sup-plementary Fig. 1) were in accordance with high-angle annular dark field scanning TEM (HAADF-STEM) observations and the amorphous character of clusters was validated by SAED analysis (Fig. 1e). EDS data showed the uniform distributions of gadolinium and calcium in the cluster aggregates (Supplementary Fig. 2a, b).

ACNC exhibited no crystalline peak in the X-ray powder diffrac-tion (XRD) pattern, even after being dispersed in aqueous solution for 6 months (Supplementary Fig. 3). The X-ray photoelectron spectro-scopy (XPS) of ACNC exhibited the characteristics peaks, corre-sponding to the oxygen (O 1s), carbon (C 1s), gadolinium (Gd 4d), and calcium (Ca 2p), respectively. In line with the previous observation of ACC[7], the peaks with binding energies of 350.8 and 347.3 eV were ascribed to Ca 2p$_{1/2}$ and Ca 2p$_{3/2}$ state, respectively. In the core-level spectrum of Gd 4d, two peaks were observed at 142.6 eV and 150.6 eV, which corresponded to Gd 4d$_{5/2}$ and Gd 4d$_{3/2}$ state, respectively (Sup-plementary Fig. 4). The ACNC was further studied by X-ray Absorption Spectroscopy (XAS) and extended X-ray absorption fine structure (EXAFS) analysis, and the Ca–O short-range environment within ACNC was revealed to closely relate to those in the ACC standard and

previously reported ACCs[10,11,23] (Fig. 1f, g, Supplementary Tables 1, 2). The size distribution of ACNC received from SAXS fitting analysis showed an average radius of 1.3 nm (Fig. 1h, i).

Thermogravimetric analysis (TGA) and differential scanning calorimeter (DSC) results for ACNC showed a loss over 20 wt% upon heating to 300 °C due to the dehydration of the ACC, indicating a high water content of ACNC. In addition, the pyrolysis of PAA occurred between 300 and 500 °C, and carbonate was decomposed above 550 °C (Fig. 1j and Supplementary Fig. 5). Based on the TGA result, the composition of ACNC included ~20% water content, 40% PAA, and 40% amorphous carbonate. The absorbance observed in TGA coupled FTIR (TG-FTIR) on ACNC at 2358 and 2322 cm$^{-1}$ above 550 °C corre-sponded to the asymmetric stretching vibration of $CO_2$ from the decomposition of carbonate (Fig. 1k). As shown in Supplementary Fig. 6, the split band at 1415 and 1454 cm$^{-1}$ in Fourier transform infrared spectroscopy (FTIR) can be assigned to the asymmetric stretch vibration of the carbonate ions in typical ACC environments[20]. A broad peak centered at 1086 cm$^{-1}$ attributed to the internal $CO_3^{2-}$ symmetric stretch was further confirmed by Raman spectroscopy (Supplementary Fig. 7).

### Contribution of Gd ion in ACC-like environment

In order to investigate the contribution of each component to the highly hydrated content and ACC-like environment, a series of control samples with varying composition were synthesized as listed in Sup-plementary Table 3, including a Gd-occluded ACC composite (ACC-Gd), PAA-stabilized ACC (ACC-PAA), amorphous gadolinium carbonate (AGC), PAA-stabilized AGC (AGC-PAA) and two carbonate-free chelates (PAA-Ca and PAA-Ca/Gd). XRD patterns and Raman spectra were per-formed to identify the amorphous phase (Supplementary Fig. 8a, b)[12]. The extended X-ray absorption fine structure (EXAFS) results further confirmed the presence of ACC-like environments (Fig. 2a, b).

In the absence of PAA, ACC-Gd showed a typical morphology for aggregated ACC nanoparticles with a uniform distribution of Gd and Ca (Supplementary Fig. 9). The $^{13}$C nuclear magnetic resonance (NMR) result of ACC-Gd showed a characteristic vibration at 168 ppm, which indicated the ACC precursor phase in ACC-Gd (Supplementary Fig. 10). FT-IR spectrum showed a consistency compared with the amorphous calcium carbonate phase, and the $v_2$ vibration shifted to lower value indicated the formation of amorphous gadolinium carbonate (AGC) phase (Supplementary Fig. 11)[24]. The oxygen (O 1s) peak of ACG's XPS result was quite similar to ACC's curve (Supplementary Fig. 12)[7]. The XPS spectrum of AGC-PAA exhibited similar profile to ACNC (Supple-mentary Fig. 13).

The transformations of ACC-Gd were investigated in different mediums (ethanol, water) to further explore how Gd affected the stability of ACC. The XRD results of samples were collected at different time points (Supplementary Fig. 14). After 6 months, calcite, aragonite, and vaterite were all contained in ACNC dispersed in ethanol. In sharp contrast, ACC-Gd was stable in ethanol over 6 months. The fresh-prepared additive-free ACC crystallized rapidly after dispersion in water, whereas ACC-Gd could maintain amorphous phase for tens of minutes. Overall, Gd should play a reasonable role in retarding the crystallization of ACC. More attractively, there was still a higher water content held in ACC-Gd as compared to pure ACC for a long time. According to the TGA result, the weight loss before 300 °C of ACC-Gd and ACC powder exposed to the air over 6 months was more than 10% and only 0.1%, respectively (Fig. 2c). Note that ACC has completely lost its hydration water under these conditions.

### Highly hydrated content of ACNC

In both TGA and volumetric Karl Fischer titration measurement results, the weight loss before 300 °C could be assigned to the loss of water, and ACNC exhibited the highest water content of calculative 20 wt% among the products with different composition (Fig. 2d). According to

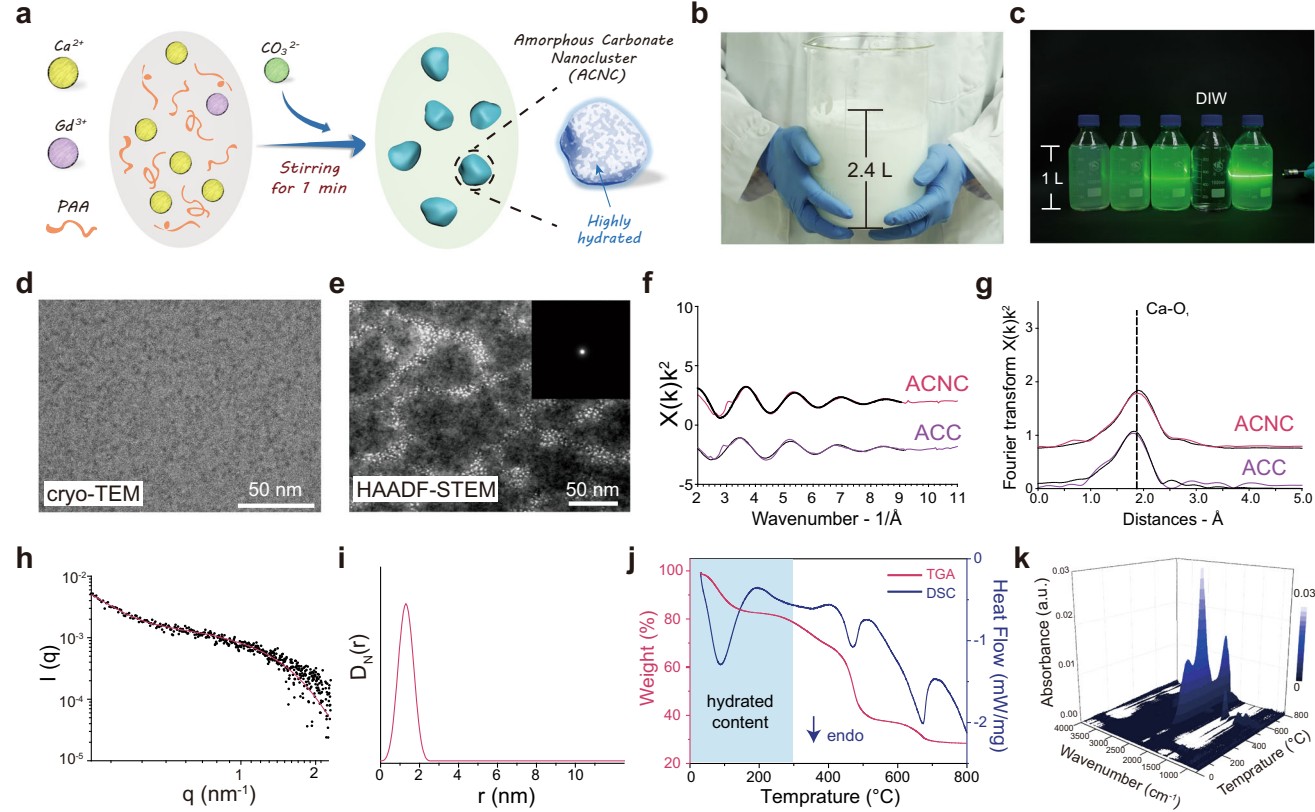

**Fig. 1 | Characterization of amorphous carbonate nanocluster (ACNC).**
**a** Scheme of the synthesis process of ACNC. **b** Digital image of the large-scale prepared ACNC with the total volume over 2 liters. **c** Digital image of Tyndall effect of four liters of ACNC dispersed in normal saline, and one bottle of deionized water was inserted (second from right). **d** cryo-TEM and **e** HAADF-STEM image of ACNC dispersed in normal saline solution. A representative image of three individual experiments is shown. The inset in (**e**) shows SAED of ACNC. The experiments were repeated three times independently. **f** $k^2$-weighted EXAFS and **g** $k^2$-weighted Fourier transform of the EXAFS for the ACNC and ACC standard. Dotted black lines are their best fits. **h** SAXS patterns of ACNC dispersed in normal saline solution. Red solid line fit sphere particles. **i** Distance distribution received by SAXS of ACNC dispersed in normal saline solution. **j** TGA of ACNC powder under an $N_2$ atmosphere with a heating rate of 10 °C min$^{-1}$. **k** TG-FTIR spectra of ACNC powder under an $N_2$ atmosphere with a heating rate of 10 °C min$^{-1}$.

previous reports, the water contents per mole of calcium in additive-free ACC and PAA-stabilized ACC were typically located in the range of 0.4−1.58 and 1.33−1.93, respectively[20,25]. In contrary, both ACC-Gd and ACC-PAA showed a higher water to Ca ratio, and the highest ratio was obtained in ACNC (water/Ca = 7.2) (Fig. 2e), indicating a synergetic enhancement of water binding by PAA and Gd in these amorphous nanoparticles. More importantly, to compare the ratio of water to Gd as listed in Fig. 2f, the hydrated content per mole gadolinium in the presence of both ACC-like environment and PAA in ACNC is more than four times that of PAA-occluded gadolinium carbonate (PAA-AGC). Compared with AGC, the presence of ACC in the Gd-ACC composite also resulted in a two-fold increase of the hydrated content per gadolinium ion.

To investigate the origin of the hydration, analytical ultra-centrifugation (AUC) was applied to determine the hydration water of ACNC via the frictional ratio and Perrin function assuming a spherical shape seen in the electron microscopy images (Supplementary Table 4). The samples show sedimentation coefficients in the order of $10^{-13}$ s, which is typical for prenucleation clusters (PNCs) that have similar sizes[26]. The ACNCs have a diameter of 1.5 nm, which is even smaller than the one found by SAXS, though still in reasonable agreement. Their molar mass of 2230 g/mol indicates that they consist of ~20 $CaCO_3$ ion pairs including the $Gd^{3+}$ ions. The determined amount of bound hydration water in solution of 8.8 mol $H_2O$ per $CaCO_3$ (and 23.0 mol $H_2O$ per $Gd^{3+}$), is considerably higher than in common ACCs (Fig. 2e, f and Supplementary Table 5). Notably, the water content of ACNC determined by AUC is in reasonable agreement

with the results from TGA, considering that the hydration was determined in wet and dry states, respectively[6].

## Stable high relaxivity of ACNC

Besides the contribution of the dopant of the gadolinium ion to the highly hydrated ACC content, a typical paramagnetic behavior was achieved in the amorphous carbonate nanoclusters as shown in the M−H curve (Fig. 3a). Both the high hydration and confinement of the gadolinium ions, the paramagnetic ACNC was conducive to increasing MRI contrast performance. The $T_1$ relaxation times of ACNC dispersed in normal saline at varying concentrations were measured using a 3.0 Tesla MR scanner. As shown in the $T_1$ map (Fig. 3b), the longitudinal relaxivity ($r_1$) of ACNC reached 37.21 mM$^{-1}$ · s$^{-1}$ (Fig. 3c), which was ten times higher than that of commercially used Gd-DTPA (3.19 mM$^{-1}$ · s$^{-1}$). Besides the contribution from the macromolecules and nanoparticles, ACNC possessed four times higher longitudinal relaxivity compared to PAA-modified amorphous gadolinium carbonate (AGC-PAA) (7.91 mM$^{-1}$·s$^{-1}$), which was attributed to the high hydration content of ACC-like environments. This high content of water may be primarily attributed to the stronger hydration ability of trivalent $Gd^{3+}$ as compared to divalent $Ca^{2+}$ ions[19], which reasonably enhanced inner-sphere and outer-sphere relaxations of the ACC-like environment occluded ACNC[14].

Here, we further evaluate the effect of the content of PAA on water content and relaxivity. We denoted the input of PAA for the ACNC product as 'n'. Using the same synthesis methods, several products with PAA inputs of '2n', 'n/2', 'n/10' (labeled ACNC$_{(2n-PAA)}$, ACNC$_{(n/2-PAA)}$,

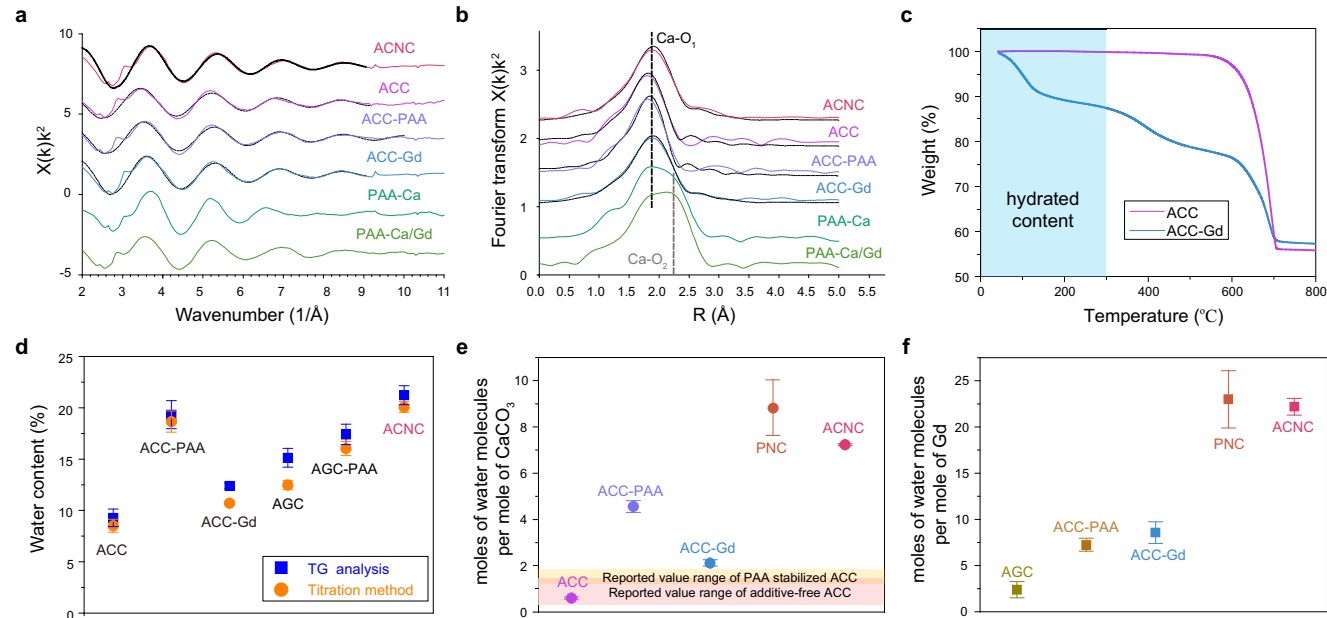

**Fig. 2 | High water content of ACNC. a** $k^2$-weighted EXAFS and **b** $k^2$-weighted Fourier transform of the EXAFS for the ACNC, ACC standard, ACC-PAA, ACC-Gd, PAA-Ca, and PAA-Ca/Gd. Dotted black lines are their best fit. **c** Thermogravimetric curve of ACC and ACC-Gd lyophilized powder isolated by deionized water and then exposed to dry air for over 6 months. The weight loss before 300 °C could be attributed to the loss of water. **d** The water contents of ACC, ACC-PAA, ACC-Gd, AGC, AGC-PAA, and ACNC according to thermogravimetric (TG) analysis and volumetric Karl Fischer titration measurements ($n = 3$ independent samples). The data show means ± SD. **e** Molar ratio of water molecules per mole of $CaCO_3$ in ACC, ACC-PAA, ACC-Gd, and ACNC in comparison with the reported value ranges of additive-free ACC (pink region) and PAA-stabilized ACC (orange region) ($n = 3$ independent samples). The data show means ± SD. **f** Molar ratio of water molecules per mole of Gd in AGC, AGC-PAA, ACC-Gd, and ACNC ($n = 3$ independent samples). The data show means ± SD.

$ACNC_{(n/10-PAA)}$) were obtained under the same experimental conditions. We found a significant increase in water content of the whole product system with PAA added (Supplementary Fig. 15). However, a 'roller coaster' effect was also detected as the molar ratio of water molecules per mole of Gd reached its peak with an input size of 'n', yet got decreased with an increased input size such as '2n' (Fig. 3d). In addition, we measured the relaxivities of these samples. Interestingly, its performance was highly consistent with that of water content per Gd in the way that relaxivity of ACNC reached the highest under the condition of 'n', yet got reduced when the amount of PAA was '2n' and 'n/2'. Specifically, the relaxivity decreased significantly when the amount of PAA was only '10/n' (Fig. 3d, e). In light of the results, it was clear that the molar ratio of water molecules per mole of Gd acted strongly on the performance of the products. These comparisons suggested that relaxivity was changing along with water content per Gd as they share a similar pathway of changing.

We further designed and fabricated two more amorphous carbonate nanoclusters with varying ratios of Gd added for comparison purpose. In the synthesis of ACNC, $mol[GdCl_3]:mol[CaCl_2]$ equals to 1:5. We also prepared two samples with initial Gd/Ca feed ratio as 1:10 and 1:2 (denoted as $ACNC_{(Gd/Ca=1:10)}$ and $ACNC_{(Gd/Ca=1:2)}$, respectively). $ACNC_{(Gd/Ca=1:10)}$ and $ACNC_{(Gd/Ca=1:2)}$ were both amorphous phases, and no crystallization peaks could be found in XRD patterns of the two products dispersed in aqueous solutions over 3 months (Supplementary Fig. 16). $ACNC_{(Gd/Ca=1:10)}$ retained a good MR performance with high relaxivity as 34.25 $mM^{-1}·s^{-1}$ when the proportion of doping Gd was low. However, when the level of incorporation of Gd was elevated, the longitudinal relaxivity ($r_1$) values of $ACNC_{(Gd/Ca=1:2)}$ decreased remarkably as 17.21 $mM^{-1}·s^{-1}$ (Fig. 3f). One explanation could be that the excessive amount of Gd may potentially perturb the highly hydrated ACC-like environment, which in turn affected the generation of ACC with high water content. In summary, the doping amount of Gd significantly affected the relaxivity of paramagnetic ACNC, ACNC found to be the best product in terms of contrast performance.

After three tests, the $r_1$ value of ACNC was calculated to be ~37.93 ± 0.63 $mM^{-1}·s^{-1}$ (Fig. 3g). In addition, $r_1$ and $r_2$ values of ACNC were measured under different magnetic fields (3.0 T and 0.5 T), and its corresponding $r_2/r_1$ ratio was calculated (Fig. 3h, i). The $r_1$ value of ACNC measured on 3.0 T was as high as 38.19 $mM^{-1}·s^{-1}$, and the corresponding $r_2$ value and $r_2/r_1$ ratio were 72.49 $mM^{-1}·s^{-1}$ and 1.90, respectively. Using a low field MR scanner system (0.5 T), the corresponding $r_1$ and $r_2$ values of ACNC were 66.37 $mM^{-1}·s^{-1}$ and 78.04 $mM^{-1}·s^{-1}$, respectively, and the $r_2/r_1$ ratio was 1.18. Based on the results of AUC measurements, the molar mass of ACNC with a diameter of 1.5 nm was 2230 g/mol. The relaxivity density of ACNC can be eventually calculated with the help of the volume and molecular weight (21.46 $mM^{-1}·s^{-1}/nm^3$, 17.01 $mM^{-1}·s^{-1}$/ kDa, respectively).

High stability is essential for gadolinium-based contrast agents, as transmetallation will lead to the release of dissociated gadolinium ion with reported toxicity such as nephrogenic systemic fibrosis[14,27]. Laurent and Muller reported the poor kinetic inertness against transmetallation for linear gadolinium-based contrast agents such as commercially used Gd-DTPA[28]. As shown in Fig. 3j, the PAA-Ca/Gd chelate exhibited an even worse inertness than Gd-DTPA after exposure to $Zn^{2+}$ in PBS for 48 h. On the contrary, ACNC demonstrably enhanced the stability against transmetallation, due to the gadolinium confinement in ACC-like environments. Furthermore, we addressed a ligand competition assay in homogeneous aqueous solution containing DTPA and ACNC. There was no observable evidence showing that relaxivity of ACNC was significantly altered, which confirmed that ACNC was not affected by the competition between Gd(III) complex and excessive DTPA-free ligand (Supplementary Fig. 17).

## Stability and biocompatibility of ACNC

The absorption spectra of Arsenazo III is generally used to detect the leakage of gadolinium ion from Gd-based nanocomposite and gadolinium chelates[29,30]. When the Arsenazo III aqueous solution was mixed with $Gd^{3+}$, pink solution turned blue due to the formation of arsenazo-

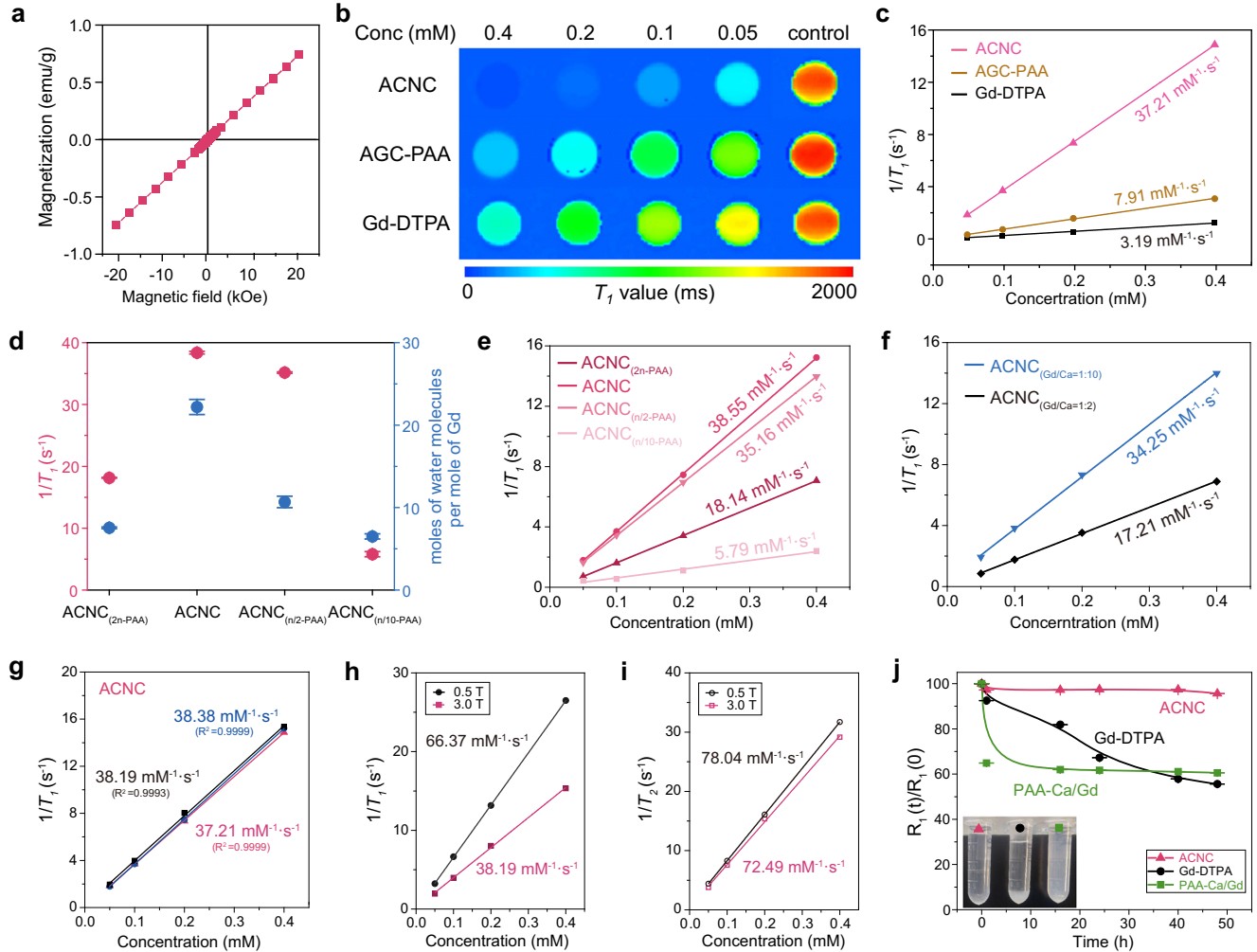

**Fig. 3 | MRI performance of ACNC. a** M–H curve of ACNC at room temperature. **b, c** $T_1$ map and $1/T_1$ versus Gd concentration curve of ACNC ($R^2 = 0.9999$), AGC-PAA ($R^2 = 0.9987$) and Gd-DTPA ($R^2 = 0.9966$) under 3.0 T. **d** Molar ratio of water molecules per mole of Gd and $r_1$ relaxivity in ACNC, ACNC$_{(2n\text{-}PAA)}$, ACNC$_{(n/2\text{-}PAA)}$ and ACNC$_{(n/10\text{-}PAA)}$ ($n = 3$ independent samples). The data show means ± SD. **e** $1/T_1$ versus Gd concentration curve of ACNC ($R^2 = 0.9999$), ACNC$_{(2n\text{-}PAA)}$ ($R^2 = 0.9999$), ACNC$_{(n/2\text{-}PAA)}$ ($R^2 = 0.9999$) and ACNC$_{(n/10\text{-}PAA)}$ ($R^2 = 0.9845$) under 3.0 T. **f** $1/T_1$ versus Gd concentration curve of ACNC$_{(Gd/Ca=1:10)}$ ($R^2 = 0.9993$) and ACNC$_{(Gd/Ca=1:2)}$ ($R^2 = 0.9996$) under 3.0 T. **g** $1/T_1$ versus Gd concentration curve of ACNC under 3.0 T ($n = 3$ independent samples). **h** $1/T_1$ versus Gd concentration curve of ACNC under 3.0 T ($R^2 = 0.9993$) and 0.5 T ($R^2 = 0.9999$). **i** $1/T_2$ versus Gd concentration curve of ACNC under 3.0 T ($R^2 = 0.9991$) and 0.5 T ($R^2 = 0.9999$). **j** Evolution of relative $R_1$ values from the zero time to each time ($R_1(t)/R_1(0)$) for ACNC, Gd-DTPA and PAA-Ca/Gd ($n = 3$ independent experiments). The data show means ± SD.

Gd$^{3+}$ complex (Fig. 4a). As shown in Fig. 4b, free gadolinium ion at a low concentration of 1 μg/mL was detectable by Arsenazo III mediated absorption spectra. However, in the normal saline dispersion of ACNC, no leakage of free gadolinium ions in 1-week dialysis was detected using this colorimetric analysis, which was further confirmed by ICP-MS (Fig. 4c). In sharp contrast to ACNC, PAA-Ca/Gd chelate showed an obvious leakage compared with ACNC, further confirming the confinement of gadolinium ions by carbonate.

It is well known that pH plays a key role in tissue and cellular homeostasis. Moreover, different cellular compartments present variety of acid-base conditions. We used PBS buffers with a pH of 6.0–6.8 to mimic the weakly acidic condition of different intracellular locations. Although the leakage of Ca ions can be observed, the leakage of Gd ion was scarcely detected with 7 days (Supplementary Fig. 18). We presume that Gd(III) within ACNC will tend to form GdPO$_4$ with phosphate in the PBS buffer due to the lower thermodynamic solubility product (Ksp) of phosphate than that of carbonate[31]. The precipitation of free Gd$^{3+}$ ions was suppressed at near neutral (pH = 6) conditions[32], which prevented leakage of Gd ions, indicating the good biosafety of ACNC under weak acid conditions. Under more acidic environmental pH (pH 4.5–5.5) simulated by acetate buffers, the

release of Gd ions slightly increases with the decrease of pH (Supplementary Fig. 18). Compared with the results in PBS environment, the improved leakage of Gd ions was ascribed to the loss of protection provided by phosphate. Therefore, we speculated that Gd(III) was difficult to leakage from ACNC and exist as ions under physiological environment.

Moreover, ACNC was dispersed in human serum at 1 mmol (Gd) /L. Meanwhile, to simulate the elevated phosphate concentrations in serum in patients with end-stage renal disease[33], the same concentration of ACNC was dispersed in human serum supplemented an additional phosphate concentration of 10 mmol/L for 15 days, followed by dialysis at different time points. No leakage of free gadolinium ions was detected in dialysates using ICP-MS and colorimetric analysis (Fig. 4d, e and Supplementary Fig. 19), indicating the low dissociation risk for ACNC in the retention period in vivo.

To determine whether ACNC cause hemolysis, ACNC at different concentrations were incubated with human blood serum at 37 °C for a hemolysis test. According to the standard, ACNC have no haemocylolysis even at a high concentration of 0.5 mg (Gd)/mL, suggesting a good blood compatibility (Fig. 4f). MTT assay was used as an evaluation of cytotoxicity by co-culturing human renal tubular epithelial cell

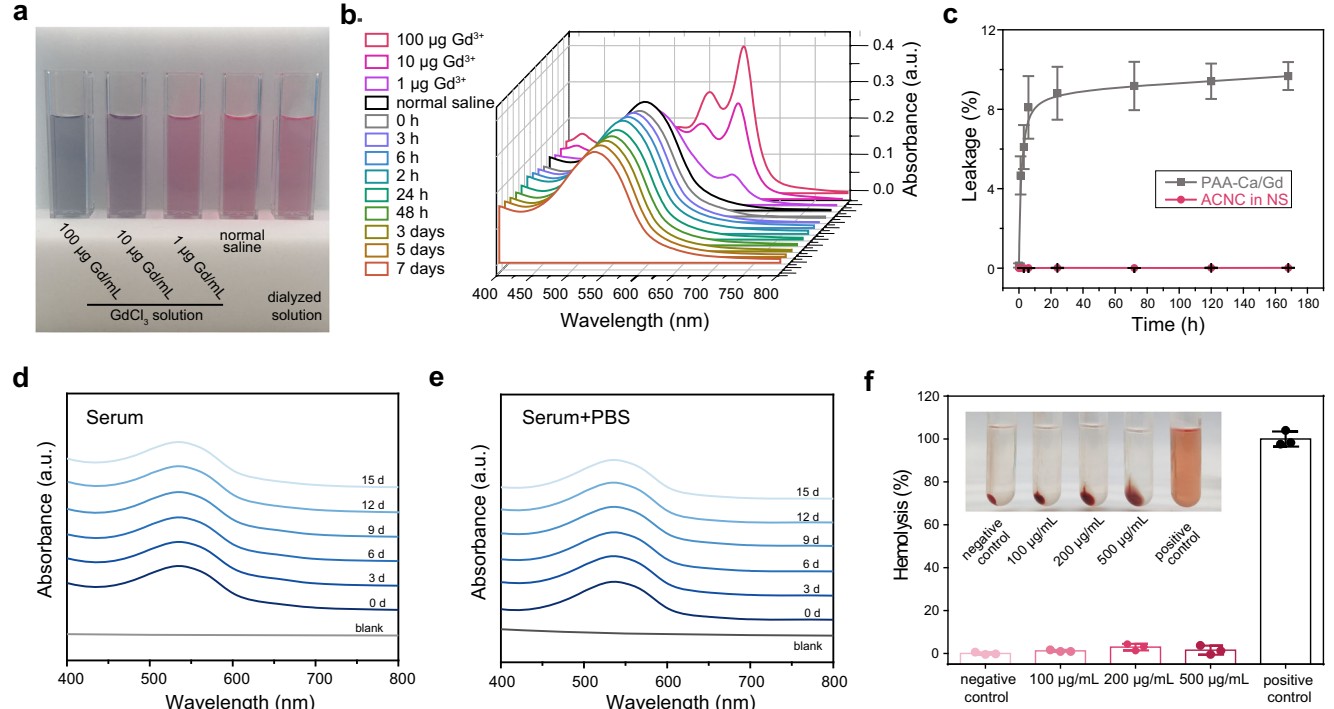

**Fig. 4 | Stability study. a** Photos of the mixtures of Arsenazo III aqueous solution with dialyzed solutions of ACNC, normal saline (NS, served as negative control), and gadolinium chloride aqueous solution at varying concentrations (1, 10, and 100 μg Gd/mL) (served as positive controls), respectively. **b** Absorption spectra of Arsenazo III-dialysate mixtures, and dialysates were collected at varying time points in 7 days to monitor the leakage of gadolinium from ACNC. No detectable leakage of gadolinium ion was monitored compared with the negative (NS) and positive (gadolinium chloride aqueous solution) controls. **c** Quantitative analysis and comparison of the leakage of free gadolinium ion from ACNC and PAA-Ca/Gd in NS by means of ICP-AES. PAA-Ca/Gd chelate showed poor inertness in NS, and ~9% gadolinium was leaked from the chelate after 7 days. In contrary, few free gadolinium ion was detected in the dialysates of ACNC (*n* = 3 independent samples). The data show means ± SD. **d, e** Absorption spectra of Arsenazo III mixture with dialyzed solutions collected at different time points from **d** human serum with dispersed ACNC and **e** human serum with dispersed ACNC and supplemented additional phosphate. **f** Blood compatibility evaluation of ACNC at varying gadolinium concentration (*n* = 3 independent samples). The data show means ± SD.

(HK-2) or human immortal keratinocyte (HaCaT) lines with ACNC for 24 h and 48 h. Little reduction of viability in cells was induced after exposure to ACNC, even at a high concentration of 0.5 mg (Gd)/mL (Supplementary Fig. 20). In addition, good cytocompatibilities of both ACNC$_{(Gd/Ca=1:10)}$ and ACNC$_{(Gd/Ca=1:2)}$ were validated (Supplementary Fig. 21). Encouraged by the results thus far, a series of subcellular compatibility studies were carried out using human renal tubular epithelial cell (HK-2). Healthy mitochondria of HK-2 cells were demonstrated by mitochondrial membrane potentials (MMP) study (Supplementary Fig. 22). The immunofluorescent TdT-mediated dUTP Nick-End Labeling (TUNEL) staining assay of HK-2 showed ACNC had no accumulated damage to the nuclear DNA of cells (Supplementary Fig. 23). An EdU (5-Ethynyl-2'-deoxyuridine) assay was performed and confirmed that ACNC had no effects on cell proliferation (Supplementary Fig. 24).

Meanwhile, histological sections of major organs were collected 2 weeks after the intravenous injection in mouse at a dose of 5 mg Gd/kg, and no evidence of pathologic change was identified in H&E stained images (Supplementary Fig. 25). The H&E staining of kidney from rabbit was also carried out, and no acute renal toxicity was observed (Supplementary Fig. 26). Three days and 3 weeks after intravenous injection of ACNC in mouse, all hematologic and biochemical parameter values were in accordance with the standard range and our control groups (Supplementary Fig. 27). To further test in the large animal beagle dog at a high dosage (9 mg Gd/kg bw), there was no obvious difference of hematologic and biochemical parameters between control and experimental groups after intravenous injection (Supplementary Fig. 28), suggesting that physiological functions were not impaired by ACNC.

## In vivo MR imaging and metabolism of ACNC

The enhancement of $T_1$ weighted MRI performance of ACNC was further confirmed by MR angiography in rat, rabbit, and beagle dog. After the bolus intravenous injection of ACNC at a low dose (3 mg/kg bodyweight), jugular, carotid, aorta, and caudal vena cava can be clearly identified (Fig. 5a–c), and the dosage is <1/5 of the typically employed value in clinic. The whole-body angiography images of rat, rabbit, and upper body of beagle dog exhibited better angiographic contrast and more details in comparison with that of the Gd-DTPA control group. Meanwhile, semiquantitative analysis clearly displayed that signal intensities in ACNC groups have a remarkable enhancement in comparison to those in Gd-DTPA groups (Supplementary Figs. 29, 30). As shown in Fig. 5d, the contrast enhancement of ACNC in vivo was further quantitatively reflected by the mean value of signal-noise ratio (SNR) on rabbit and beagle dog, respectively[34].

Recently, clinical translation of nanosized biomaterials attracted great attention, promising the development of novel and specific nanomedical tools for diagnosis and therapy[16,35–38]. The majority of administered nanomaterials were sequestered by liver and spleen, which thus become the major biological barriers for translating nanomedicines[39]. To avoid the potential risk in vivo, renal clearance is the desirable and preferred excretion route for medical agents for minimal catabolism and body exposure. However, compared to small molecular contrast agents, nanosized MR contrast agents, especially the FDA-approved nanosized iron oxide injections suffer from their poor renal excretion as a result of their large size distribution (>20 nm)[40]. The liver is the primary or secondary target of transmission for nanoparticles with access to the circulatory system, resulting in inevitable accumulation of nanoparticles in the liver.

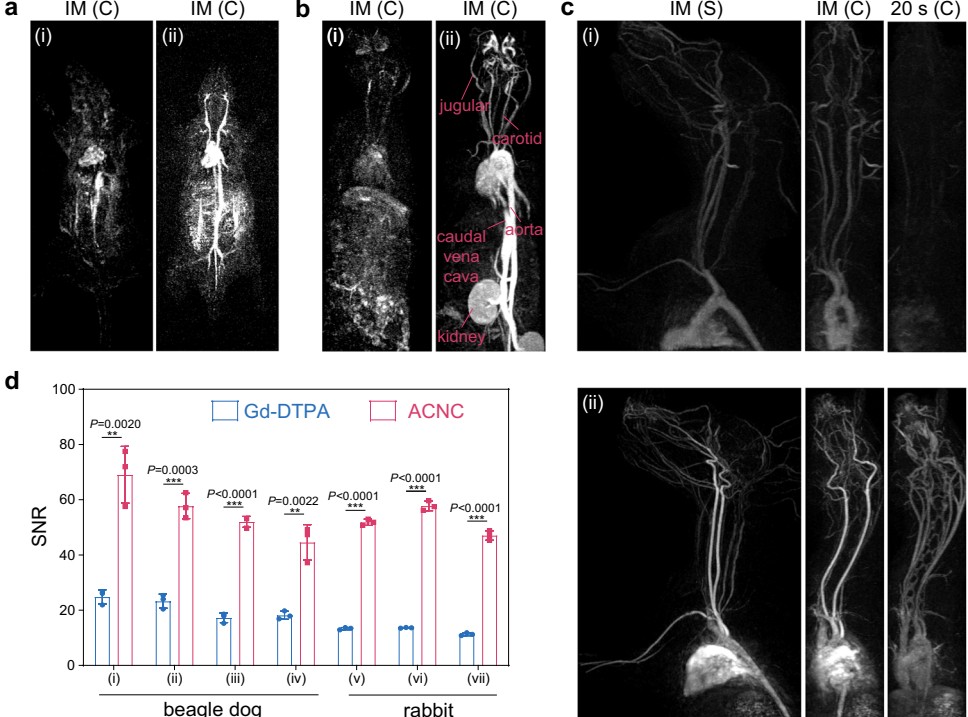

**Fig. 5 | In vivo MR angiography. a, b** Contrast-enhanced MR angiography (MRA) images of the whole body on **a** rat and **b** rabbit immediately after the bolus injection of (i) Gd-DTPA and (ii) ACNC. **c** MRA images of upper body on beagle dog at immediately (IM) and 20 s after the bolus injection of (i) Gd-DTPA and (ii) ACNC, respectively. (C) and (S) represents coronal plane and sagittal plane respectively. **d** Quantitative measurement of SNR of ROIs in (i) brachiocephalic trunk artery, (ii) left subclavian artery, (iii) left common carotid artery, (iv) right branch of olfactory artery of beagle dog, and that in (v) descending aorta, (vi) aortic arch, (vii) ascending aorta of rabbit, respectively. The time for quantitative measurement of SNR of ROIs was in the "IM" period ($n = 3$ independent experiments). The data show means ± SD. $P$ was calculated using two-tailed Student's $t$-test (**$P < 0.01$, ***$P < 0.001$).

Nanoparticles detained in the liver could be eliminated from the liver via hepatobiliary clearance[41]. Besides the conventional elimination from liver (Supplementary Fig. 31), the quick renal clearance of ACNC from blood vessels was observed in the MRA images after intravenous injection (Supplementary Fig. 32). Moreover, the bladder of beagle dog was also brightened within 20 min in the $T_{1w}$ image (Supplementary Fig. 33). In addition, as shown in the blood concentration-time curve in mouse and beagle dog measured by ICP-AES, it could be effectively cleared from blood vessels in 6 h and there is rarely any residual content of gadolinium after 24 h (Fig. 6a, b and Supplementary Table 6). In the collected rat urine after the intravenous injection of ACNC, the content of gadolinium was detected by ICP-AES and demonstrated a renal clearance efficiency of ~13% ID at 24 h (Supplementary Fig. 34), which is comparable to that of gold nanoclusters with similar diameters[42]. Abundant SAED amorphous cluster aggregates could be observed in TEM images of dialyzed urine (Fig. 6c), and EDS mapping revealed a matching distribution of gadolinium, calcium, carbon, and oxide elements in these aggregates corresponding to ACNC (Fig. 6d, e). Physicochemical and physiological stability in synergy with low injection dosage and partial clearance via kidney led to in vivo biocompatibility and potential translational ability of these gadolinium-based amorphous carbonate clusters.

## Discussion

A number of gadolinium-based contrast agents have been designed and approved worldwide for $T_1$ MR imaging[14]. As a result of a high density of free electrons in the valence band and versatile functionalization to interact with biomolecules in vivo, inorganic nanomaterials serving as contrast agents in various imaging modalities are anticipated to achieve the clinical translation of nanomedicine[15,43].

Therefore, the scientists studied indefatigably including diverse fabrication methods and clearing pharmacokinetics in order to investigate the translatability of Gd-based inorganic nanoparticles[16]. In consideration of the hazard originated from the release of free Gd ions, a specific aim was to reduce the dose of injected nanoparticles via the enhanced performance.

The hydration level plays a significant role in determining the contrasting performance of a MR contrast agent[14]. Unfortunately, the hydration content of Gd-based nanocrystals is limited by traditional high-temperature synthesis processes[15]. Worth mentioning is that the high moisture content, including interior water and deeply located structural water is the most distinctive feature of ACC[7-11]. However, the potential application of unstable hydrated ACC is largely ignored. Interestingly, the ionic radius of the gadolinium ion is very close to that of the calcium ion, implying a possible interaction between gadolinium ions and calcium ions that can be exploited by us.

In summary, our study confirms mutual effects between the paramagnetic lanthanide gadolinium ion and amorphous calcium carbonate, which is beneficial to maximizing the water content in the obtained amorphous composite nanoclusters. The material is synthesized through a facile one-pot process at room temperature, enabling the large-scale and cost-effective production. Importantly, this high water content contributes to the transparent MRI contrasting enhancement of gadolinium-based nanoclusters. In combination with their low toxicity, partial renal clearance and easy potential for mass production, our work enables further identification of the biomedical potential of ACC composites, and we anticipate that it could lead to a next generation of more efficient diagnostic agents on the basis of amorphous nanoclusters in the future.

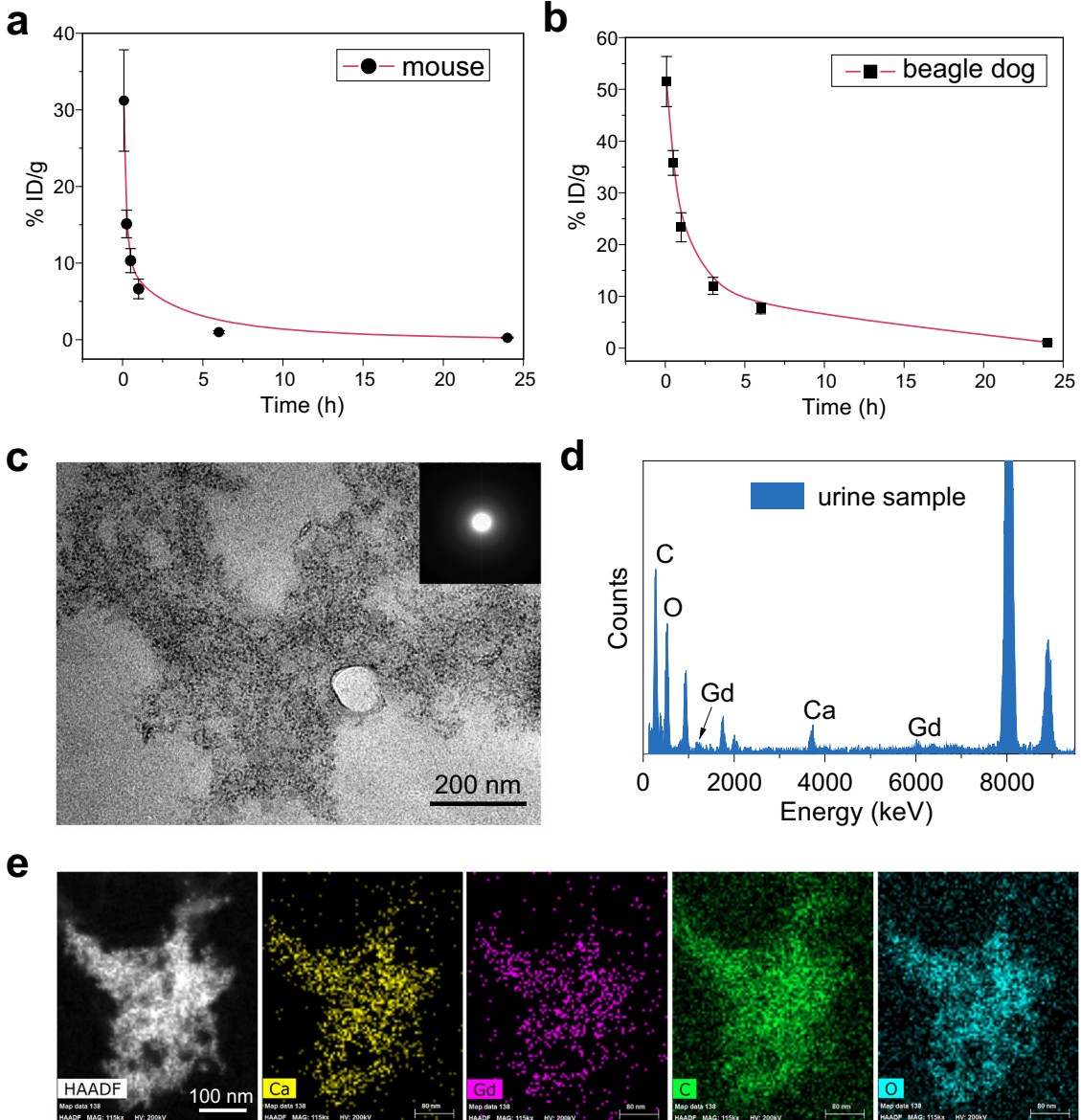

**Fig. 6 | Metabolism of ACNC. a, b** The time-dependent distribution of ACNC in plasma of **a** mice and **b** beagle dogs within 24 h ($n = 5$ biologically independent animals). The data show means ± SD. **c** TEM, **d** EDS results, and **e** HAADF-STEM image and EDS mapping of ACNC observed in rat urine collected 12 h post-injection. A representative image of three individual experiments is shown. Inset of **c** is relative SAED pattern. The experiments were repeated three times independently.

## Methods

### Materials

Anhydrous calcium chloride, anhydrous sodium carbonate, and ethyl alcohol were purchased from Sinopharm Chemical Reagent Co. Ltd (Shanghai). Gadolinium chloride hexahydrate (GdCl$_3$·6H$_2$O, 99.95%) were purchased from Yutai QingDa Fine Chemical Co., Ltd. (Shandong, China). Poly(acrylic acid) (PAA, Mw ≈ 1800) was purchased from Aldrich. All reagents were used as received without further purification. The commercially used Gd-DTPA was produced by Guangdong Consun Pharmaceutical Group, China (specification: 20 mL).

### Sample preparation

In a typical procedure to synthesize additive-free amorphous calcium carbonate (ACC), 10 mL Na$_2$CO$_3$ aqueous solution (0.1 mol·L$^{-1}$, 10 mL) was poured into calcium chloride (0.1 mol·L$^{-1}$, 10 mL) aqueous solutions under vigorous magnetic stirring for 15 s. The aqueous suspension after adding 20 mL ethanol were centrifuged at 850 × $g$ for 2 min

immediately, then the precipitates were washed by ethanol, centrifuged for 5 min at 5000 × $g$ twice.

In a typical procedure to synthesize ACNC, anhydrous calcium chloride (CaCl$_2$, 0.1 mol·L$^{-1}$), gadolinium chloride hexahydrate (GdCl$_3$, 0.02 mol·L$^{-1}$), and PAA (0.45 g) were dissolved in 10 mL DIW with magnetic stirring. Anhydrous sodium carbonate (Na$_2$CO$_3$, 0.1 mol·L$^{-1}$, 10 mL) was poured into the above solution with vigorous magnetic stirring for 1 min. 20 mL of ethanol was added to terminate the reaction. The mixture was immediately centrifuged for 2 min at 850 × $g$, and precipitates were re-suspended by DIW, centrifuged for 5 min at 5000 × $g$ twice. For liter-scale synthesis, the reaction volume was scaled up 50 times and proceeded under robust mechanical agitation.

For the synthesis of ACC stabilized by PAA (ACC-PAA), 0.45 g PAA was used in the synthesis process without the addition of gadolinium salt, and the product was washed by ethanol. For the preparation of ACC-Gd, PAA was absent in the synthesis process and anhydrous CaCl$_2$ (0.1 mol·L$^{-1}$) and GdCl$_3$ (0.02 mol·L$^{-1}$) were mixed in 10 mL DIW. Then the obtained ACC-Gd was washed by ethanol.

In a typical procedure to synthesize amorphous gadolinium carbonate (AGC) and AGC-PAA, $Na_2CO_3$ (0.1 mol·L$^{-1}$, 10 mL) was poured into 10 mL $GdCl_3$ aqueous solution (0.067 mol·L$^{-1}$, mol[$GdCl_3$]:mol[$Na_2CO_3$] = 2:3) under vigorous magnetic stirring for 15 s, then 20 mL ethanol was poured into the aqueous suspension. After centrifuged at $850 \times g$ for 2 min immediately, the precipitates were washed by ethanol, centrifuged for 5 min at $5000 \times g$ twice. In addition, in the presence of 0.45 g PAA in the $GdCl_3$ aqueous solution, AGC-PAA was prepared in the same procedure.

## Characterization

Transmission electron microscopy (TEM) were carried out on H7700 (Hitachi, Japan) with an acceleration voltage of 100 KV. Cryo-transmission electron microscopy (cryo-TEM) was performed on Tecnai F20 Transmission Cryo-Electron Microscopy. High-angle annular dark field scanning TEM (HAADF-STEM), energy dispersive spectrometer (EDS), and EDS mapping were performed on a field emission high-resolution transmission electron microscope (FEI, Talos F200X). X-ray powder diffraction (XRD) patterns were performed on a rotating anode X-ray diffractometer (SmartLab$^{TM}$ 9 kW). X-ray photoelectron spectroscopy (XPS) was performed on a photoelectron spectrometer (ESCALAB, 250Xi, Thermo Fisher, USA). $^{13}C$ nuclear magnetic resonance (NMR) was performed on 400 MHz WB Solid-State Nuclear Magnetic Resonance Spectrometer (Bruker AVANCE III 400 WB). FT-IR spectra were recorded with a Thermo Nicolet 8700 FT-IR spectrometer at room temperature. Raman spectra were recorded using a Horiba LabRAM high-resolution (HR) Evolution Raman spectrometer. The light source of the microscope was transferred to a diode laser (780 nm). The spectra were scanned for $3 \times 100$ s[44,45]. Thermogravimetric analysis (TGA), differential scanning calorimeter (DSC), and thermogravimetry analysis coupled with Fourier transform infrared spectroscopy (TG-FTIR) were recorded at a heating rate of 10 K min$^{-1}$ under nitrogen flow on a TGA thermal analyzer (NETZSCH STA 449F3). Magnetic investigation was carried on a superconducting quantum interface device (SQUID) magnetometer (Quantum Design SQUID-VSM). ICP-AES were performed on an Optima 7300 DV instrument. UV-vis spectra were measured on a Shimadzu UV-2600 spectrophotometer at room temperature.

## Small-angle X-ray scattering (SAXS)

Small-angle X-ray scattering (SAXS) analysis were performed on a SAXSpoint 2.0 (Anton Paar). The size distribution function as determined by SAXS was evaluated using GIFT (Generalized indirect Fourier transformation) software with precondition of homogeneous spheres weighted by number. It is calculated using the following formulas[46]:

$$I(q) = \int_0^{R_{max}} R^6 \cdot D_N(R) \cdot \left( 3 \cdot \frac{\sin(qR) - qR\cos(qR)}{(qR)^3} \right)^2 \cdot dr \quad (1)$$

## EXAFS sample preparation, measurement, and data fitting

X-ray absorption spectroscopy (XAS) measurements at the calcium K-edge (4.0381 keV) were undertaken at the Elettra synchrotron Italy, operating at an energy of 2 GeV and current of 300 mA. Samples were ground in an agate mortar, diluted with graphene powder, and compacted into thin pellets. An optimal sample thickness of ~300 μm and optimal calcium concentration after optimal dilution with graphene powder was prepared for each sample including ACNC, ACC-Gd, and ACC standards. Three full scans per sample of the X-ray absorption near edge structure (XANES) and extended X-ray absorption fine structure (EXAFS) were collected in transmission mode using a FMB-OXFORD ion chamber detector, in steps of 5 eV in the pre-edge region (from 3738.43 to 4028.53 eV) and 0.2 eV in the edge region (from 4061.51 to 4061.71 eV), gradually increasing to 2.6 eV in the post-edge

region (4061.71 to 4589.30 eV) up to $k = 13$. Alignment, energy calibration and deglitching were performed using built-in features of the Athena software package[47]. The first coordination shell around calcium (Ca–O) was fitted by generating a single scattering pathway of Ca–O based on crystallographic data of calcite. Analysis of single scattering pathways was undertaken from 1.3 to 2.53 Å in R space on Fourier transformed $k^2$-weigthed data from 3 to 9.1 Å. Standards errors associated with the EXAFS data fitting over the k-range used here are 15% for the first-shell coordination number, 0.03 Å for the radial distance, and 15% for the Debye-Waller factors.

Independent data points were determined by Stern's rule defining the fundamental limitation to the amount of information that can be determined by XAFS. According to Stern's rule, the number of parameters ($N_{par}$) used for the fit should be strictly inferior to the number of independent ($N_{ind}$) parameters defined as $2\Delta k\Delta R/\pi + 2$, where $\Delta k$ and $\Delta R$ are the fitted range in $k$ and $R$ space, respectively[48,49]. If $N_{ind} < N_{par}$, the model cannot be taken as a proof of the coordination environment because the data set is underdetermined by the level of complexity of the model. In this case, $N_{ind} = 7.5$ for ACC standard, ACC-Gd, and ACNC sample, $N_{ind} = 8.16$ for ACC-PAA and $N_{par} = 4$ as such than $N_{ind} > N_{par}$. The multi-component model yielding the best fit of the experimental data is considered as reasonable and can thus be considered as a likely representation to characterize the calcium coordination environment. Linear combination was performed on the XANES and near-EXAFS region (from 30 Å$^{-1}$ before to 80 Å$^{-1}$ after edge jump). Samples ACC-Gd, ACNC, and PAA-Gd/Ca were compared to standards samples ACC-PAA and PAA-Ca.

## Analytical ultracentrifugation (AUC)

The AUC measurements were performed on a modified Optima XL-A (Beckman Coulter, Palo Alto, CA, United States) using an absorbance optics and an advanced Rayleigh interference optics developed by Nanolytics (https://www.nanolytics-instruments.de/interference_optics_aida). 12-mm path length double-sector titanium centerpieces with saphire windows (Nanolytics, Potsdam, Germany) were used for all experiments. For ACNC sample measurement, the original ACNC sample was pre-sedimented on a UNIVERSAL 320 Hettich centrifuge (Hettich, Tuttlingen, Germany) for 20 min at 9 000 rpm, corresponding to a centrifugal force at $6000 \times g$. 360 μL of pretreated ACNC solution was used in the sample sector and 360 μL of a 1:175 diluted (with 0.154 M NaCl) pre-reaction PAA-Ca/Gd solution (aqueous mixture including PAA, calcium chloride and gadolinium chloride) in the reference sector. For PAA sample measurement as reference, 10 mg PAA (1800 Da) were dissolved in 1 mL of 0.154 M NaCl solution. All samples were investigated at 20 °C and 60,000 rpm, corresponding to a centrifugal force up to $290,000 \times g$.

Sedimentation velocity data were evaluated with Sedfit and UltraScanIII. In Sedfit (Schuck, P. P. SEDFIT version 16.1c. http://analyticalultracentrifugation.com/download.htm), the ls-g*(s) and continuous c(s,ff$_0$) models were used. For calculation of the sedimentation coefficient distributions g(s) with the least-squares-g*(s) model[50], data were fitted with a Tikhonov-Phillips regularization using a confidence level (F-ratio) of 0.683 and a resolution of 100 grid points. For determination of the sedimenting particle density[51] with the c(s,ff$_0$) analysis[52], data were fitted with a maximum entropy regularization using a confidence level (F-ratio) of 0.683 and a resolution of 100 grid points in s and 10–20 grid points in $f/f_0$. The 2D c(s,ff$_0$) distributions were plotted using MATLAB software version R2017b, 64 bit from MathWorks. For calculation of the sample density from 2D distributions, a MATLAB mask script developed by Quy Khac Ong was used (Institute of Materials, École Polytechnique Fédérale de Lausanne (EPFL), Station 12, 1015 Lausanne, Switzerland. E-mail: quy.ong@epfl.ch). For characterization of anisotropy, UltraScanIII (Demeler, B. UltraScan version 4.0, release 2783. A Comprehensive Data Analysis Software Package for Analytical Ultracentrifugation Experiments. The University of Lethbridge,

Department of Chemistry and Biochemistry. http://www.ultrascan3.aucsolutions.com/download.php) was used for performing the two-dimensional spectrum analysis (2DSA)[53]. The 2DSA-Monte Carlo (MC) analyses were performed with 50 iterations on a supercomputer.

The function defining the shape, flexibility, and degree of solvation (by water, salt ions, and any other solvent molecules) of the macromolecule is the *Perrin translational frictional function, P* (see following equation). This degree of water association is termed the hydration of the solute, $\delta$, and is defined as the mass in grams of associated solvent per gram of anhydrous solute[54].

$$P = \frac{f}{f_0}\left(\frac{\bar{v}}{\bar{v}_s}\right)^{1/3} = \frac{f}{f_0}\left(\frac{\bar{v}}{\bar{v} + \delta\,\bar{v}_{H2O}}\right)^{1/3} \qquad (2)$$

where $f/f_0$ is the frictional ratio (which is the dimensionless ratio of the observed translational frictional coefficient $f$ to that of an equivalent spherical molecule of the same anhydrous mass and density $f_0$), $\bar{v}$ is the partial specific volume (cm³/g) of the solute and $\bar{v}_s$ is the hydrated specific volume (the volume occupied by the solute and associated solvent per unit mass of the anhydrous solute) and $\bar{v}_{H2O}$ = specific volume of water given as:

$$\bar{v}_s = \bar{v} + \delta\,\bar{v}_{H2O} \qquad (3)$$

If all excess friction is due to the hydration and the solute is a sphere, then $P$ is 1[55] and the hydration can be calculated as:

$$\delta = \frac{\left[\left(\frac{f}{f_0}\right)^3 - 1\right]\bar{v}}{\bar{v}_{H2O}} = \left[\left(\frac{f}{f_0}\right)^3 - 1\right]\bar{v}\,\rho_{H2O} \qquad (4)$$

### Hydrated content analysis
The moles of water molecules per mole of $CaCO_3$ was thus calculated according to the result of inductively coupled plasma atomic emission spectroscopy (ICP-AES), TG analysis, and volumetric Karl Fischer titration measurement. Titration measurement was performed on a Karl Fischer moisture titrator (V10S, volumetric KF titrator, Mettler Toledo). The hydrated content was assigned to be the weight loss starting from the endothermic peak to 300 °C, while the content of $Ca^{2+}$ and $Gd^{3+}$ ions were determined by ICP-AES. The molar ratio between water and $Ca^{2+}$ ions in ACC can thus be calculated according to previous reports[20,56,57].

### Leakage studies of $Gd^{3+}$ ion
The detection of leakage of gadolinium ion was carried out using a method reported previously[29,30]. ACNC were dispersed in a series of simulative physiological environment including normal saline, human serum, and human serum supplemented an additional phosphate concentration of 10 mmol/L. ACNC was dispersed with a final concentration of 1 mmol (Gd)/L. The dialysis of ACNC was performed at 37 °C for 7 days using a dialysis bag (MWCO 1000 Da). The concentration of gadolinium in dialysates collected at different time points was measured by both Arsenazo III mediated chromogenic assay and ICP-AES. In the chromogenic assay, the collected dialysates were mixed with Arsenazo III (0.05 mM) dissolved in chloroacetic acid-sodium hydroxide buffer solution (pH 2.8), and the absorption at 658 nm was detected. Normal saline and $GdCl_3$ solution were used as negative and positive controls, respectively.

### Transmetallation study
ACNC, Gd-DTPA, and PAA-Ca/Gd samples were freshly prepared. At $t = 0$, each sample (2.5 mM Gd) and $ZnCl_2$ aqueous solution (250 mM, 20 μL) were mixed in a 2 mL phosphate buffer. Then, 1 mL mixed solution was contained in a chromatographic bottle for measurement.

The measurements were carried out at 37 °C under a 0.5 T NMI20-Analyst NMR Analyzing and Imaging system. TR/TE = 100/5.6 ms. Longitudinal relaxation times were measured at different points in time[28]. The relaxation rate at $t = 0$ (denoted as $R_1(0)$) was calculated by the formula $R_1 = (1/T_1)$. The relaxation rates at other time points were all respectively calculated and recorded as corresponding $R_1(t)$, for the purpose of assessing the 2-day transmetallation through monitoring the ratio of $R_1(t)/R_1(0)$.

### Ligand competition assay
1 mL ACNC aqueous solution (5 mM Gd) was added into 1 mL DTPA solution (5 mM) and the homogeneous solution was taken for measurement. Compared with the commercial Gd-DTPA, the concentration of Gd-DTPA solutions was identical to ACNC solutions as 5 mM Gd. The measurements and calculations were consistent with transmetallation study.

### Degradation experiments
1 mL ACNC aqueous solution (1 mg (Gd)/mL) was introduced into dialysis bags (MWCO: 1000 KD). The bags were then placed in 50 mL PBS or acetate buffers with different pH and incubated with shaking at 50 rpm at 37 °C. We collected all the surrounding PBS or acetate solutions at each time interval for analysis and then replaced with fresh 50 mL PBS or acetate buffers. In order to prevent possible interference which the free metal ions generated from degradation precipitate with the phosphate ions in the PBS buffer, 1 mL of freshly prepared chloroazotic acid ($HNO_3$/HCl = 3:1) was added to the collected surrounding PBS or acetate solution, leading to a strong acid environment (pH < 3.0) of the mixed solution. Ultimately, the concentrations of Gd ion in surrounding solutions were determined by ICP-AES.

### In vitro biocompatibility studies
Human renal tubular epithelial (HK-2) cell and human immortal keratinocyte (HaCaT) cell line were cultured in Dulbecco's modified Eagle's medium. Cell viability was performed by standard MTT method. Briefly, cells were plated at the density of about $1 \times 10^4$ cells per well in 96-well plate, and incubated for 12 h. Then DMEM culture medium with ACNC were added, and cells were exposed to ACNC at various concentrations for 24 h and 48 h. Then 10 μL of MTT solution (5 mg/mL in PBS) was added to each well for additional 4 h incubation at 37 °C. After removing the medium, 150 μL of DMSO was added to dissolve the formed formazan crystals in each well, and the optical density of the solution was measured at 570 nm using Microplate Reader (BioTek Instruments, USA). HK-2 cell was used for mitochondrial membrane potentials (MMP) assay, TdT-mediated dUTP Nick-End Labeling (TUNEL) assay, and 5-Ethynyl-2′-deoxyuridine (EdU) assay. Mitochondrial membrane potential assay kit with J-aggregates (JC-1) (Beyotime, Shanghai, China), one-step TUNEL apoptosis assay kit (Beyotime, Shanghai, China), and EdU cell proliferation kit with Alexa Fluor 555 (Beyotime, Shanghai, China) were employed.

### Blood studies
The blood for the experiment was professionally collected from volunteers by medical professionals in blood transfusion department of the first affiliated hospital of Anhui medical university. The blood studies were carried out in accordance with the protocols approved by the Ethics Committee of University of Science and Technology of China and Anhui Medical University (license number: 20170267, 20170268).

### Animal studies
Specific pathogen Free (SPF) BALB/c mice (male, 6 weeks), New Zealand rabbit (male, 2 kg), and Beagle dog (male, 6 kg) were purchased from Anhui Medical University. Animal studies were performed in accordance with the recommendations in the Guide for the Care and Use of Laboratory Animals of the National Institutes of Health. The

in vivo compatibility studies and in vivo MRI studies were under protocols approved by the Institutional Animal Care and Use Committee (IACUC) of Anhui Medical University (LLSC20170299, LLSC20170300) and The First Affiliated Hospital of University of Science and Technology of China (2021-N(A)-041).

## In vivo biocompatibility studies

ACNC normal saline solutions were intravenously injected in mice (via tail vein) at a dosage of 5 mg Gd/kg bodyweight and Beagle dog (via hindlimb vein) at a dosage of 9 mg Gd/kg bodyweight. For evaluation of blood index and biochemical index, mice were sacrificed by anesthetics for 3 days and 3 weeks after intravenous injection, respectively. Blood of Beagle dogs were collected every two weeks via hindlimb vein. Blood samples were collected by the anticoagulant blood collection tubes and separating gel tubes. The examination of blood index and biochemical index were carried out on Sysmex XE2100 and Vitros 5600, respectively.

## In vivo metabolic study

ACNC normal saline solutions were intravenously injected into twenty mice (male, 20 g) at 3 mg Gd/kg bodyweight (0.5 mg Gd/mL, 120 μL) via rapid manual injection for a simulated bolus injection study. These mice were randomly divided into four groups, and various organs were resected from the mice in each group 1, 7, 15, and 30 days after intravenous injection, respectively. The urine samples were collected at 12 h after *i.v.* injection of ACNC on rat at 5 mg Gd/kg bodyweight, followed by dialysis at 25 °C. After 24 h, the dialysates in dialysis bags (MWCO = 1000 Da) were collected for further characterization.

## MRI acquisition and analysis

ACNC and Gd-DTPA dispersed in normal saline at gradient concentrations in 5 mL microtubes were placed on a support immersed in $NiSO_4$ solution. The concentrations of gadolinium were measured by ICP-AES. MR $T_1$ map was acquired using an inversion recovery sequence on a 3.0 Tesla MR scanner (Trio Tim, Siemens) equipped with a head coil. Imaging parameters were as follows: repetition time (TR) = 4000 ms, echo time (TE) = 14 ms, inversion time (TI) from 25 to 3500 ms (TI included 25, 50, 75, 100, 150, 200, 250, 300, 350, 400, 500, 600, 800, 1000, 1500, 2000, 2500, 3000, 3500 ms), field of vision (FOV) = 220 × 144 mm². The longitudinal relaxivity ($r_1$) was calculated as follow according to an equation reported previously with a concentration gradient from 0.05 to 0.4 mM of gadolinium ion. The inverse of the relaxation time ($1/T_1$, s⁻¹) was plotted against the concentration of gadolinium, and the slope of the plot was the relaxivity of the agent (mM⁻¹·s⁻¹)[58,59]. The measurements on a low field MR scanner system were performed using a 0.5 T LF-NMR instrument at 32 °C provided by Suzhou Niumag Analytical Instrument Corporation.

## In vivo MR angiography analysis

After complete intravenous anesthesia, animals were fixed onto different supports with suitable size. Then, ACNC dispersed in normal saline or Gd-DTPA diluted by normal saline were injected intravenously. Rat was injected from caudal vein using an indwelling needle at a dose of 2.5 mg Gd/kg. Bolus injection on rabbit and beagle dog were performed at a dose of 3 mg Gd/kg from forelimb veins by mechanical high-pressure injector (Optistar LE, Mallinckrodt, USA). FLASH 3D sequence was employed to collect angiographic data. The detailed parameters of MR angiography of rat were as follows: A knee coil was used. TR = 3.95 ms, TE = 1.9 ms, FOV: 210 × 280 mm². The time of acquisition (TA) was 24.62 s. Flip angle (FA) was 20. Acquisition matrix was 288p * 512. Acquisition number and number of averages were both 1. The detailed parameters of MR angiography of rabbit were as follows: A local head and neck coil were used. TR = 3.6 ms, TE = 1.65 ms, FOV: 210 × 280 mm². The time of acquisition (TA) was 23.4 s. Flip angle (FA) was 18. Acquisition matrix was 288p * 512. Acquisition number and number of

averages were both 1. The detailed parameters of MR angiography of beagle dog were as follows: A local head and neck coil, spine coil, and radiofrequency body transmit coil were used. TR = 3.19 ms, TE = 1.28 ms, FOV: 240 × 320 mm². The time of acquisition (TA) was 20.74 s. Flip angle (FA) was 16. Acquisition matrix was 288p * 512. Acquisition number and number of averages were both 1. The images were processed firstly by Siemens Syngo MR workstation.

Signal-noise ratio (SNR) was calculated in a single image (κ) based on two separate regions of interest (ROIs). One in the vessel of interest ($ROI_{vessel}$) were measured by placing a signal at the same ROI in the same slice both in Gd-DTPA and ACNC group, which was recorded as the mean signal intensities of vessel ($S_{vessel}$). One in the image background ($ROI_{background}$) was located in an artifact-free homogeneous area, which was defined as the background signal ($S_{background}$)[34]. SNR corresponds to the ratio between $S_{vessel}$ and $S_{background}$ was calculated using Eq. (5):

$$\text{SNR}_{stdv}(\kappa) = \frac{S_{vessel}}{S_{background}} = \frac{\underset{r \in ROI_{vessel}}{\text{mean}}(S_N(r,\kappa))}{\sqrt{\frac{2}{4-\pi}}\underset{r \in ROI_{background}}{\text{stddev}}(S_N(r,\kappa))} = \frac{m_{vessel}}{\sqrt{\frac{2}{4-\pi}}S_{background}}$$
(5)

Three ROIs were selected in descending aorta, aortic arch, and ascending aorta on rabbit, and four regions in brachiocephalic trunk artery, left subclavian artery, left common carotid artery and right branch of olfactory artery were measured on beagle dog, respectively.

## Statistical analysis

All data were expressed as mean ± SD. Two-tailed Student's *t*-test was used for statistical analysis of between two groups comparison, and one-way ANOVA was used for statistical analysis of among multiple groups comparison. A value of $P < 0.05$ was considered statistically difference (*$P < 0.05$, **$P < 0.01$, ***$P < 0.001$). Statistical significance was determined using SPSS statistics 17.

## Reporting summary

Further information on research design is available in the Nature Research Reporting Summary linked to this article.

## Data availability

The authors declare that the data generated in this study are provided in the paper and Supplementary Information, or available from the corresponding authors on request.

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

## Acknowledgements

This work was supported by the National Natural Science Foundation of China (Grants 51732011 and U1932213 to S.H.Y.; 22122502 and 51972090 to Y.L.; 51702309 to L.D.; 81801831 to H.Q.W.), the National Key Research and Development Program of China (Grants 2021YFA0715700 and 2018YFE0202201) to S.H.Y., the Fundamental Research Funds for the Central Universities (WK9110000062 to Y.J.X.; WK2060190056 to L.D.), the University Synergy Innovation Program of Anhui Province (Grant GXXT-2019-028) to S.H.Y., Science and Technology Major Project of Anhui Province (201903a05020003) to S.H.Y., the Natural Science Foundation of Anhui Province (2008085J06 to Y.L.; 1708085ME114 to Y.J.X.), the China Postdoctoral Science Foundation (2015M570540) to L.D., the Major/Innovative Program of Development Foundation of Hefei Center for Physical Science and Technology (2016FXZY005) to Y.L. The authors would like to thank Mei Sun, Yan-Wei Ding, Cheng-Min Wang, Yu-Song Wang, Guan-Yin Gao, Han-Bao Chong, Yu-Feng Meng, Yang-Yi Liu in University of Science and Technology of China, Hao Ding in Suzhou Niumag Analytical Instrument Corporation, Yong-Hong Song, Wen-Shu Wu in Hefei University of Technology, Hai-Shen Qian in Anhui Medical University, He Chen, Li Zhang and Hui Wang in The First Affiliated Hospital of Anhui Medical University, Kun Liu in Xiamen University, Duo An in Cornell University, and Ye-Ping Li from Anton Paar China for useful assistance of this manuscript, and Luca Olivi, Giuliana Aquilanti and Simone Pollastri in the XAFS beamline at Elettra Synchrotron for their help. D.G. is a professor of the Leibniz Universität Hannover. J. A. is financed within the framework of the SFB 1214 (Collaborative Research Center funded by the German Research Foundation, DFG, project A02). H.C. thanks the particle analysis center of the SFB 1214 for the AUC equipment.

## Author contributions

S.H.Y. and Y.L. conceived the idea, designed the experiments, and supervised the research, L.D., C.S., Y.L., Y.Z., L.B.M., D.G., H.C., R.R., Y.D.W., and F.L. carried out the synthetic experiment and analysis. D.G., J.A., H.C., and L.D. worked on the EXAFS measurement and analysis. L.D., Y.Z., H.L.G., Z.P., H.Q.W., X.Y., and F.L. processed the biocompatible evaluation. L.D., Y.J.X., Y.D.W., Y.L., H.L.G., Z.P., X.Y., F.L., and C.S. worked on the animal experiments, L.D., Y.J.X., Y.L., F.L., and C.S. investigated the MRI performance, L.D., Y.J.X., C.S., D.G., Y.L., H.C., and S.H.Y. wrote the paper, all authors discussed the results and commented on the manuscript.

## Competing interests

The authors declare no competing interests.

## Additional information

[1]Department of Chemistry, Institute of Biomimetic Materials & Chemistry, Anhui Engineering Laboratory of Biomimetic Materials, Division of Nanomaterials & Chemistry, Hefei National Research Center for Physical Sciences at the Microscale, Institute of Energy, Hefei Comprehensive National Science Center, University of Science and Technology of China, Hefei 230026, China. [2]The Cancer Hospital of the University of Chinese Academy of Sciences (Zhejiang Cancer Hospital), Institute of Basic Medicine and Cancer (IBMC), Chinese Academy of Sciences, Hangzhou, Zhejiang 310022, China. [3]Anhui Province Key Laboratory of Advanced Catalytic Materials and Reaction Engineering, School of Chemistry and Chemical Engineering, Hefei University of Technology, Hefei 230009, China. [4]Department of Radiology, Anhui Provincial Hospital, The First Affiliated Hospital of University of Science and Technology of China, Hefei 230001, China. [5]Institute of Inorganic Chemistry, Leibniz Universität Hannover, Callinstr. 9, 30167 Hannover, Germany. [6]Physical Chemistry, Department of Chemistry, University of Konstanz, Universitätsstr. 10, Konstanz D-78457, Germany. [7]Scientist - Center for X-ray Analytics, Empa - Swiss Federal Laboratories for Materials Science and Technology, Lerchenfeldstrasse 5, 9014 St. Gallen, Switzerland. [8]Department of Blood Transfusion, The First Affiliated Hospital of Anhui Medical University, Hefei 230022, China. [9]These authors contributed equally: Liang Dong, Yun-Jun Xu, Cong Sui. ✉e-mail: yanglu@hfut.edu.cn; helmut.coelfen@uni-konstanz.de; shyu@ustc.edu.cn

