## [Peer Review File · Nature Communications]

Highly hydrated paramagnetic amorphous calcium carbonate nanoclusters as a MRI contrast agentREVIEWER COMMENTS

Reviewer #1 (Remarks to the Author):

This manuscript describes the development of highly hydrated paramagnetic amorphous calcium carbonate nanoclusters (ACNC) as a superior MRI contrast agent. ACNC shows high relaxivity, good MR angiography in different animal models and rapid renal clearance properties. This work provides new insight for the design of MRI contrast agents with clinical translational potential. However, before it is considered for publication, the authors need to address the following questions.

1. The authors claim that the high water content of ACNC contributes to its much-enhanced MRI contrast efficiency compared with Gd-DTPA. However, there is no direct evidence to prove this point. A control group, such as ACNC with different particle size, PAA and water content, should be provided and evaluate the relationship between its water content and relaxivity.
2. Amorphous calcium carbonate is unstable in aqueous solution and will degrade under acidic conditions (such as in lysosomes and endosomes). Gd³⁺ can cause nephrogenic systemic fibrosis (NSF) in patients with kidney disfunctions. So, it is important to evaluate the degradation and safety of ACNC under weak acid conditions. How Gd ions are released after ACNC degradation under weakly acidic conditions?
3. Arsenazo III mediated chromogenic assay was used to detection of leakage of Gd ion in this work. Would direct mixing of ACNC and Arsenazo III under simulative physiological environment be a reasonable way to evaluate its stability?
4. The authors claim that ACNC can be effectively cleared by the kidneys. However, Fig.S21 shows that Gd³⁺ is mainly distributed in the liver. Even after 1 day and 7 days, the content of Gd³⁺ in the liver is much higher than other organs. The author should explain the reason for this phenomenon. Gd³⁺ distribution in different organs should be performed.
5. Major hematologic and biochemical indices of liver and kidney assessment after injection of ACNC in animals should be performed.

Reviewer #2 (Remarks to the Author):

In this work, the authors developed ultrafine paramagnetic amorphous carbonate nanoclusters (ACNC) in the presence of both gadolinium occluded highly hydrated ACC-like environment and poly(acrylic acid). Basically, the nanoclusters were well designed and performed with convincing data. Furthermore, the authors demonstrated the MR imaging potential of this nanoclusters in vitro and in vivo by some careful studies. The work is well organized and results are clearly presented. Therefore, I recommend this manuscript to be published in Nature Communications. There are still some questions might be considered in this manuscript.

1. The XPS characterization should also be provided to analyze the distinction of binding energy of Gd-O between these three samples (AGC, AGC-PAA and ACNC). Similarly, the binding energy of Ca-O needs to be characterized as same above.
2. The scale bars should be mentioned in the captions, such as Figure 1 d/e, Figure 6 c/e.
3. The error bar in Figure 3e-3g & Figure 4c should be provided.
4. The semiquantitative analysis in Figure 5a&5c and Supplementary Figure S22&S23 should be provided.
5. The renal clearance of ACNC in mice should also be characterized.

Reviewer #3 (Remarks to the Author):

The authors reported interesting finding for potential MRI agents, however, there are some important issues, especially biological safety, I don't think this manuscript can be published.

Major issues:

1. Because of the amorphous state, the samples should be non-homogeneous microscopically. Therefore, there is no evidence to show how much sample is actually involved in the tests, especially those involved cells. High/low local concentrations of the samples could give a bias to the conclusion, and the reproducibility of tissue/cell-related experiments might be poor.
 2. Not like other nanoclusters applied as MRI contrast agents, there is no indication of "relaxivity density" (defined as [relaxivity per unit]/[unit volume or molecular weight]) or "relaxivity per Gd" (commonly used to judge Gd-chelates). Those parameters are more specific and more applicable than an overall relaxivity value, especially for the macromolecules studied here.
 3. Because the structure of ACNC is not uniform and not clearly reported, it might be difficult to propose further modifications on this material. Not like nanoparticles or complexes which can be coated or conjugated with vectors for targeted therapy/imaging, there seems little room for the amorphous materials.
 4. The stability tests are not comprehensive. It is not surprising that ACNC did not leak free Gd(III) ions obviously in aqueous solution or in in vitro tests. However, there at least should be a competition test between ACNC and other Gd(III) chelators. For examples, the competition between a new Gd(III) complex and excessive DTPA free ligand is a classical way to compare the stability.
- Besides, although Fig. 3G is described as results for stability tests (line 165). Its picture caption and axis labels are unclear and there is no information on how this experiment was conducted.
5. It could be difficult to precisely control the administration amount in practical uses (minimal dosage) because the Gd content in a tiny amount of amorphous materials could vary largely and then the performance will be quite different.
 6. Mentioned in the manuscript is that most ACC are instable in aqueous solution which limits their applications, but the authors report ACNC to be biocompatible. It is necessary and will be impactful to explain in detail how the added Gd(III) could dramatically change the physical properties.
 7. Other cell lines for MTT may be expected. Only one cell line for biocompatibility seems not convincing enough. In addition, different incubation times may also be required in this experiment.
 8. For the confocal imaging (Fig. S14 and Fig. S15, p10, line190-192), Why are the incubation times for these two experiments different? In addition, results with longer incubation time will be more convincing for this study.
 9. For the clearance study, in Fig.S21, the amounts of ACNC remaining in the liver and spleen in mice after even 30 days were not negligible and could be harmful.
 10. In the nanocluster characterisation part (p5), although there is no exact formula for the nanocluster, the doping percentage of gadolinium in ACNC is still expected to be mentioned for the following analysis and comparison. How will the amount of added Gd(III) salts affect the overall performance (relaxivity, toxicity, stability)? There is no discussion on it.

Reviewer #1 (Remarks to the Author):

This manuscript describes the development of highly hydrated paramagnetic amorphous calcium
carbonate nanoclusters (ACNC) as a superior MRI contrast agent. ACNC shows high relaxivity, good MR
angiography in different animal models and rapid renal clearance properties. This work provides new
insight for the design of MRI contrast agents with clinical translational potential. However, before it is
considered for publication, the authors need to address the following questions.

1. The authors claim that the high water content of ACNC contributes to its much-enhanced MRI contrast
efficiency compared with Gd-DTPA. However, there is no direct evidence to prove this point. A control
group, such as ACNC with different particle size, PAA and water content, should be provided and evaluate
the relationship between its water content and relaxivity.

** Thanks for your valuable comments.

Based on our previous research on water content of ACNC, the addition of PAA is able to enhance
the water content of the product. Here, to further evaluate the effect of the content of PAA on water content
and relaxivity, different amount of PAA was added to synthesize a series of amorphous calcium carbonate
nanoclusters. We denoted the input of PAA for the ACNC product as 'n'. Using the same synthesis methods,
several products with PAA inputs of '2n', 'n/2', 'n/10' (labeled ACNC_(2n-PAA), ACNC_(n/2-PAA), ACNC_{(n/10-}
PAA)) were obtained under the same experimental conditions.

We found a significant increase in water content of the whole product system with PAA added (Figure
1.1a). However, a 'roller coaster' effect was also detected as the molar ratio of water molecules per mole
of Gd (water content per Gd) reached its peak with an input size of 'n', yet got decreased with an increased
input size such as '2n' (Figure 1.1b). In addition, we measured the relaxivities of these samples.
Interestingly, its performance was highly consistent with that of water content per Gd in the way that

relaxivity of ACNC reached the highest under the condition of ‘n’, yet got reduced when the amount of
 PAA was ‘2n’ and ‘n/2’. Specifically, the relaxivity decreased significantly when the amount of PAA was
 only ‘10/n’ (Figure 1.1b, c).

In light of the results, it is clear that water content per Gd acts strongly on the performance of the
 products. These comparisons suggest that relaxivity is changing along with water content per Gd as they
 share a similar pathway of changing.

**Figure 1.1 a** TGA of ACNC, ACNC_(2n-PAA), ACNC_(n/2-PAA) and ACNC_(n/10-PAA) powder under an N₂
 atmosphere with a heating rate of 10 °C min⁻¹. **b** Molar ratio of water molecules per mole of Gd and r_1
 relaxivity in ACNC, ACNC_(2n-PAA), ACNC_(n/2-PAA) and ACNC_(n/10-PAA) (n = 3). **c** 1/T₁ versus Gd
 concentration curve of ACNC (R² = 0.9999), ACNC_(2n-PAA) (R² = 0.9999), ACNC_(n/2-PAA) (R² = 0.9999)
 and ACNC_(n/10-PAA) (R² = 0.9845) under 3.0 T.

 2. Amorphous calcium carbonate is unstable in aqueous solution and will degrade under acidic conditions
 (such as in lysosomes and endosomes). Gd³⁺ can cause nephrogenic systemic fibrosis (NSF) in patients
 with kidney disfunctions. So, it is important to evaluate the degradation and safety of ACNC under weak
 acid conditions. How Gd ions are released after ACNC degradation under weakly acidic conditions?

** Thanks for your valuable comments.

The safe and effective clinical use of Gd-based complexes as MRI contrast agents was getting
sustained attention. The administration of gadolinium-based MRI contrast agents (GBCA) have been
postulated to result in nephrogenic systemic fibrosis (NSF) since 2006 [Ref. 1. *Nephrol. Dial. Transplant.*
2006, 21, 1104-1108; Ref. 2. *J. Am. Soc. Nephrol.* 2006, 17, 2359-2362]. In addition, the U.S. Food and
Drug Administration (FDA) recently warned that GBCA for MRI was concerned with gadolinium
remaining in patients' bodies, including the brain, for months to years after receiving these drugs.
Therefore, it is significant and essential to adequately explore the degradation and safety of ACNC in
different physiological environment.

It is well known that pH plays a key role in tissue and cellular homeostasis. What is worth mentioning,
although the extracellular pH of normal tissue and blood maintains at 7.4, the solid tumor
microenvironment holds an acidic pH of 6.0-7.0 [Ref. 3. *Chem. Soc. Rev.* 2017, 46, 3830-3852]. In
addition, different cellular compartments present variety of acid-base conditions. Endosome's acidic pH
stays within 5.5-6.5, while lysosome's falls within 4.5-5.5 [Ref. 4. *Nat. Mater.* 2013, 13, 204-212; Ref. 5.
*Nano Today.* 2015, 10, 656-670]. We used PBS buffers with a pH of 6.0-6.8 to mimic the weakly acidic
condition of extracellular tumor microenvironment and different intracellular locations. Meanwhile,
acetate buffers with a pH of 4.5-5.5 was used to mimic the lower pH environment.

The degradation experiments were performed in vitro as follows: 1 mL ACNC aqueous solution (1
18 mg (Gd)/mL) was introduced into dialysis bags (MWCO: 1,000 KD). The bags were then placed in 50 mL
PBS or acetate buffers with different pH and incubated with shaking at 50 rpm at 37 °C. We collected all
the surrounding PBS or acetate solutions at each time interval for analysis and then replaced with fresh 50
21 mL PBS or acetate buffers.

In order to prevent possible interference which the free metal ions generated from degradation
precipitate with the phosphate ions in the PBS buffer, 1 mL of freshly prepared chloroazotic acid
($\text{HNO}_3/\text{HCl} = 3:1$) was added to the collected surrounding PBS or acetate solution, leading to a strong
acid environment ($\text{pH} < 3.0$) of the mixed solution. Ultimately, the concentrations of Gd ion in surrounding
solutions were determined by ICP-AES.

**Figure 1.2** Cumulative leakage of free calcium and gadolinium ions from ACNC in phosphate buffers (pH
6.8, 6.5, and 6.0) and acetate buffers (pH 5.5, 5.0, and 4.5) at 37 °C within 7 days.

Based on the results as shown in Figure 1.2, although an obvious leakage of Ca ions can be observed
at a pH range of 6.0-6.8, the leakage of Gd ion was scarcely detected with 7 days. We presume that Gd(III)
within ACNC will tend to form GdPO_4 with phosphate in the PBS buffer due to the lower thermodynamic
solubility product (K_{sp}) of phosphate than that of carbonate [Ref. 6. Chem. Soc. Rev. 2018, 47, 357-403;
Ref. 7. J. Mater. Chem. B 2014, 2, 8378-8389]. Furthermore, the precipitation of free Gd^{3+} ions is

suppressed at acidic (pH = 1) or near neutral (pH = 6) conditions [Ref. 8. *J. Alloys Compd.* 2011, 509,
4160-4166; Ref. 9. *Chem.-Eur. J.* 2005, 11, 2183-2195], which prevented leakage of Gd ions, indicating
the good biosafety of ACNC under weak acid conditions. Under more acidic environmental pH (pH 4.5-
5.5), the release of Gd ions slightly increases with the decrease of pH. Compared with the results in PBS
environment, the improved leakage of Gd ions was ascribed to the loss of protection provided by
phosphate. Therefore, we speculated that Gd(III) was difficult to leakage from ACNC and exist as ions
under physiological environment.

References for response letter:

[Ref. 1.] Grobner, T. Gadolinium - a specific trigger for the development of nephrogenic fibrosing
dermatopathy and nephrogenic systemic fibrosis? *Nephrol. Dial. Transplant.* 21, 1104-1108 (2006).

[Ref. 2.] Marckmann, P. et al. Nephrogenic systemic fibrosis: Suspected causative role of gadodiamide
used for contrast-enhanced magnetic resonance imaging. *J. Am. Soc. Nephrol.* 17, 2359-2362 (2006).

[Ref. 3.] Dai, Y., Xu, C., Sun, X. & Chen, X. Nanoparticle design strategies for enhanced anticancer
therapy by exploiting the tumour microenvironment. *Chem. Soc. Rev.* 46, 3830-3852 (2017).

[Ref. 4.] Wang, Y. G. et al. A nanoparticle-based strategy for the imaging of a broad range of tumours by
nonlinear amplification of microenvironment signals. *Nat. Mater.* 13, 204-212 (2013).

[Ref. 5.] Cheng, R., Meng, F. H., Deng, C. & Zhong, Z. Y. Bioresponsive polymeric nanotherapeutics for
targeted cancer chemotherapy. *Nano Today* 10, 656-670 (2015).

[Ref. 6.] Qi, C., Lin, J., Fu, L. H. & Huang, P. Calcium-based biomaterials for diagnosis, treatment, and
theranostics. *Chem. Soc. Rev.* 47, 357-403 (2018).

[Ref. 7.] Qi, C. et al. Synthesis, characterization and applications of calcium carbonate/fructose 1,6-
bisphosphate composite nanospheres and carbonated hydroxyapatite porous nanospheres. *J. Mater. Chem.*
**B 2**, 8378-8389 (2014).

[Ref. 8.] Wang, H. Li, G. S., Guan, X. F. & Li, L. P. Synthesis and conductivity of GdPO₄ nanorods:
Impacts of particle size and Ca²⁺ doping. *J Alloys Compd.* **509**, 4160-4166 (2011).

[Ref. 9.] Yan, R. X., Sun, X. M., Wang, X., Peng, Q. & Li, Y. D. Crystal structures, anisotropic growth,
and optical properties: Controlled synthesis of lanthanide orthophosphate one-dimensional nanomaterials.
*Chem.-Eur. J.* **11**, 2183-2195 (2005).

3. Arsenazo III mediated chromogenic assay was used to detection of leakage of Gd ion in this work.
Would direct mixing of ACNC and Arsenazo III under simulative physiological environment be a
reasonable way to evaluate its stability?

** Thanks for your valuable comments.

As a generally used method to detect the leakage of gadolinium ion, the absorption spectra of
Arsenazo III has been successfully employed in gadolinium chelates in human serum and urine [Ref. 10.
*J. Pharmaceut. Biomed.* 1995, 13, 927-932], Gd-doped NaYF₄ Nanocrystals [Ref. 11. *Adv. Funct. Mater.*
**2009**, 19, 853-859; Ref. 12. *Biomaterials* 2018, 158, 74-85.] and Gd-doped LDH nanocomposite [Ref. 13.
*Biomaterials* 2013, 34, 3390-3401], which was used for ACNC in our manuscript. When the Arsenazo III
aqueous solution was mixed with Gd³⁺, pink solution turned blue due to the formation of Arsenazo-Gd³⁺
complex (Figure 1.3a). Furthermore, the formation of arsenazo-Gd³⁺ complex at low concentrations of
Gd ion can be characterized by the absorption peak at 658 nm in UV-Vis spectra. As shown in Figure 1.3b,

free gadolinium ion at 1 $\mu\text{g}/\text{mL}$ was detectable by arsenazo III, while the concentration of ACNC dialysis
solution was about 2.5 (Gd) mg/mL . So, the limit of detection (LOD) was superior to 0.1%, which was
sensitive to detect the leaked gadolinium ion.

**Figure 1.3 a** Photos of the mixtures of Arsenazo III aqueous solution with dialyzed solution, normal saline
(NS, served as negative control) and gadolinium chloride aqueous solution at different concentrations
(served as positive controls), respectively. **b** Absorption spectra of arsenazo III mixture with dialyzed
solutions collected at several different time points in comparison with negative and positive controls. No
detectable leakage of gadolinium ion was monitored in the dialysis period.

In order to assess the accuracy of spectroscopy method and its LOD, we measured quantitatively the
possible leakage of gadolinium ion from the ACNC using Inductively Coupled Plasma Mass Spectrometry
(ICP-MS), whose sensitivity against gadolinium was transparently higher than that of absorption spectra
of arsenazo III.

In serum stability research, ACNC was dispersed in human serum with a final concentration of 1
16 mmol (Gd) /L . Meanwhile, to simulate the elevated phosphate concentrations in serum in patients with
17 end-stage renal disease, the same concentration of ACNC was dispersed in human serum supplemented
an additional phosphate concentration of 10 mmol/L .

The co-incubated serum samples were loaded into dialysis bags with the MWCO of 1000 at different
time points during 15 days, then the dialysis bags were immersed into PBS and PBS with an additional
phosphate concentration of 10 mmol/L for 1 day at 37 °C. In summary, no characteristic absorption
corresponding to arsenazo-Gd³⁺ complex was detected in two kinds of serum samples at any time (Figure
1.4). This study was a valid evaluation to investigate the ACNC stability in serum at normal and elevated
phosphate concentrations. Similar with the previous absorption spectra of arsenazo III result, no free
gadolinium was detected in dialysate by ICP-MS (Figure 1.5), which further confirmed the confinement
of gadolinium ions by carbonate.

**Figure 1.4 a** Absorption spectra of arsenazo III mixture with dialyzed solutions collected at several
different time points from human serum with dispersed ACNC. No detectable leakage of gadolinium ion
was monitored in the dialysis period. **b** Absorption spectra of arsenazo III mixture with dialyzed solutions
collected at several different time points from human serum with dispersed ACNC and supplemented
additional phosphate. No detectable leakage of gadolinium ion was monitored in the dialysis period.

 **Figure 1.5** Analysis of the leakage of free gadolinium ion from ACNC dispersed in human serum and
 human serum with supplemented additional phosphate by means of ICP-MS.

 **References for response letter:**

[Ref. 10.] Hvattum, E., Normann, P. T., Jamieson, G. C., Lai, J. J. & Skotland, T. Detection and quantitation
 of gadolinium chelates in human serum and urine by high-performance liquid-chromatography and
 postcolumn derivatization of gadolinium with Arsenazo-III. *J. Pharmaceut. Biomed.* 13, 927-932 (1995).

[Ref. 11.] Kumar, R., Nyk, M., Ohulchansky, T. Y., Flask, C. A. & Prasad, P. N. Combined Optical and
 MR Bioimaging Using Rare Earth Ion Doped NaYF₄ Nanocrystals. *Adv. Funct. Mater.* 19, 853-859 (2009).

[Ref. 12.] Liu, K., et al. Stable gadolinium based nanoscale lyophilized injection for enhanced MR
 angiography with efficient renal clearance. *Biomaterials* 158, 74-85 (2018).

[Ref. 13.] Wang, L. J., et al. A Gd-doped Mg-Al-LDH/Au nanocomposite for CT/MR bimodal imagings
 and simultaneous drug delivery, *Biomaterials* 34, 3390-3401 (2013).

4. The authors claim that ACNC can be effectively cleared by the kidneys. However, Fig.S21 shows that
Gd³⁺ is mainly distributed in the liver. Even after 1 day and 7 days, the content of Gd³⁺ in the liver is
much higher than other organs. The author should explain the reason for this phenomenon. Gd³⁺
distribution in different organs should be performed.

** Thanks for your valuable comments.

Renal elimination and hepatobiliary elimination are two main mechanisms executed to clear
intravenously administered nanoparticles from circulation, and then eliminate them from the body [Ref.
14. *J. Controlled Release* 2016, 240, 332-348]. In the early renal elimination stage (2 h after injection),
due to the increasing nanoparticle core density, the amount of nanoparticles excreted into the urine
decreased exponentially [Ref. 15. *Angew. Chem. Int. Ed.* 2016, 55, 16039-16043; Ref. 16. *Nat. Rev.*
*Mater.* 2018, 3, 358-374]. The liver is the primary or secondary target of transmission for nanoparticles
with access to the circulatory system, resulting in inevitable accumulation of nanoparticles in the liver
[Ref. 17. *Adv. Mater.* 2022, 34, 2106456].

Furthermore, nanoparticle size is the key factors affecting filtration efficiency, as glomerular filtration
slits in the kidneys are responsible for filtration efficiency. Since the threshold for filtration size of the
glomerular capillary wall is usually 6-8 nm (5.5 nm for metal-based nanoparticles), decreasing the particle
size of inorganic nanoparticles is the first step in improving renal clearance of these particles [Ref. 16.
*Nat. Rev. Mater.* 2018, 3, 358-374; Ref. 18. *ACS Nano* 2015, 9, 6655-6674].

In the biodistribution study mentioned in our previous manuscript (Figure 1.6a), a noticeable
distribution of Gd element in liver was observed. The main reason for this result is that ACNC was
administered with simulated bolus injections (~1 mL/s) to mimic the rapid clinical medication of MR
contrast agent via high-pressure syringe that was administrated in our in vivo studies. ACNC would form

aggregates with larger size due to this injection method, which is difficult to quickly clear from the kidney.
Given this, we performed another pilot evaluation of ACNC biodistribution in vivo with a relatively slow
injection speed ($\sim 500 \mu\text{L/s}$). The result showed that intravenous injection with lower speed could
significantly decrease the accumulation of ACNC in the liver and spleen, which provides valuable guiding
experience for the pre-clinical administration (Figure 1.6b).

**Figure 1.6** The time dependent distribution of ACNC in organs of mice after **a** rapid manual injection or
**b** normal manual injection of ACNC.

References for response letter:

[Ref. 14.] Zhang, Y.N., Poon, W., Tavares, A.J., McGilvray, I.D. & Chan, W.C.W. Nanoparticle-liver
interactions: Cellular uptake and hepatobiliary elimination. *J. Controlled Release* 240, 332-348 (2016).

[Ref. 15.] Tang, S.H., et al. Tailoring renal clearance and tumor targeting of ultrasmall metal nanoparticles
with particle density. *Angew. Chem. Int. Ed.* 55, 16039-16043 (2016).

[Ref. 16.] Du, B. J., Yu, M. X. & Zheng, J. Transport and interactions of nanoparticles in the kidneys. *Nat.*
*Rev. Mater.* 3, 358-374 (2018).

[Ref. 17.] Li, J. J., Chen, C. Y. & Xia, T. Understanding Nanomaterial-Liver Interactions to Facilitate the
Development of Safer Nanoapplications, *Adv. Mater.* 34, 2106456 (2022).

[Ref. 18.] Yu, M. X. & Zheng, J. Clearance Pathways and Tumor Targeting of Imaging Nanoparticles. *ACS*
*Nano* 9, 6655-6674 (2015).

5. Major hematologic and biochemical indices of liver and kidney assessment after injection of ACNC in
animals should be performed.

** Thanks for your valuable comments.

We add the value range of the normal parameters according to previous reports (Reference ranges of
hematology data of healthy Balb/c mice were obtained from Charles River Laboratories
(<http://www.criver.com/>), [Ref. 19. *Nat. Commun.* 2016, 7, 13193]. Meanwhile, the control normal range
explored by our group on 30 healthy Balb/c mice (Figure 1.7). According to these standard data and the
measured indices in control group, there was no obvious difference in the blood biochemical indexes in
experimental groups with i.v. injection of ACNC (5 mg/mL). Moreover, according to this valuable
suggestion and comment, we supplied further compatibility evaluation of ACNC on beagle dog (Figure
1.8). This dosage (9 mg/kg) is three times higher than that for MRA. These data were supplied in this
revised manuscript.

1
 2 **Figure 1.7** Major hematologic and biochemical indices of liver and kidney were examined 3 days and 3
 3 weeks after the intravenous injection of ACNC in BALB/c mice (n = 5), and the intravenous injection of
 4 normal saline was set as control. Major indices were listed, including white blood cells (WBC), red blood
 cells (RBC), hemoglobin (HGB), hematocrit (HCT), mean corpuscular volume (MCV), mean corpuscular
 hemoglobin (MCH), mean corpuscular hemoglobin concentration (MCHC), and platelets (PLT), alanine
 transaminase (ALT), aspartate transaminase (AST), alkaline phosphatase (ALP), blood urea nitrogen
 (BUN), and creatinine (CRE). Reference ranges of hematologic and biochemical indices of healthy Balb/c
 mice (the blue labelled region) were obtained from Charles River Laboratories (<http://www.criver.com/>).
 Meanwhile, the reference normal range from 30 healthy Balb/c mice in our group was marked by pink.

1
2 **Figure 1.8** Blood biochemical examination and blood routine examination in beagle dog at a high dose (9
3 mg/kg bw). Major hematologic and biochemical indices of liver and kidney were examined every 2 weeks
after the intravenous injection of ACNC in beagle dogs (n = 5), and the intravenous injection of normal
saline was set as control. Major indices were listed, including white blood cells (WBC), red blood cells
(RBC), hemoglobin (HGB), hematocrit (HCT), mean corpuscular volume (MCV), mean corpuscular
hemoglobin (MCH), mean corpuscular hemoglobin concentration (MCHC), and platelets (PLT), alanine
transaminase (ALT), aspartate transaminase (AST), alkaline phosphatase (ALP), blood urea nitrogen
(BUN), and creatinine (CRE).

**References for response letter:**
[Ref. 19.] Chen, Q., et al. Photothermal therapy with immune-adjuvant nanoparticles together with
checkpoint blockade for effective cancer immunotherapy. *Nat. Commun.* 7, 13193 (2016).

Reviewer #2 (Remarks to the Author):

In this work, the authors developed ultrafine paramagnetic amorphous carbonate nanoclusters (ACNC) in
the presence of both gadolinium occluded highly hydrated ACC-like environment and poly(acrylic acid).
Basically, the nanoclusters were well designed and performed with convincing data. Furthermore, the
authors demonstrated the MR imaging potential of this nanoclusters in vitro and in vivo by some careful
studies. The work is well organized and results are clearly presented. Therefore, I recommend this
manuscript to be published in Nature Communications. There are still some questions might be considered
in this manuscript.

1. The XPS characterization should also be provided to analyze the distinction of binding energy of Gd-O
between these three samples (AGC, AGC-PAA and ACNC). Similarly, the binding energy of Ca-O needs
to be characterized as same above.

** Thanks for your valuable comments.

On the basis of your suggestion, ACNC, AGC-PAA and AGC were analyzed by X-ray photoelectron
spectroscopy (XPS). The overall survey spectra of ACNC exhibited the predominant characteristics peaks,
corresponding to the oxygen (O 1s), carbon (C 1s), gadolinium (Gd 4d), and calcium (Ca 2p), respectively
(Figure 2.1). The O 1s peak of ACNC could be deconvoluted into four peaks at 531.1 eV, 531.9 eV, 532.6
17 eV, and 533.4 eV, corresponding to metal oxide (M-O), carbonate-like species (CO_3^{2-}), hydroxyl groups
(H_2O), and C=O bonds (C=O) [Ref. 1. Appl. Surf. Sci. 2013, 264, 864-871]. In the core-level spectrum of
Gd 4d, two peaks were observed at 142.6 eV and 150.6 eV, which corresponded to Gd $4d_{5/2}$ and Gd $4d_{3/2}$
state, respectively [Ref. 2. Talanta 2022, 238, 123028; Ref. 3. ACS Appl. Mater. Interfaces 2017, 9, 8142-
8150]. In line with the previous observation of amorphous calcium carbonate (ACC) [Ref. 4. J. Am. Chem.

Soc. 2018, 140, 14289-14299], the peaks with binding energies of 350.8 and 347.3 eV were ascribed to
Ca 2p_{1/2} and Ca 2p_{3/2} state, respectively.

Figure 2.1 a XPS survey spectrum of ACNC, and the corresponding spectrum of b O 1s, c Gd 4d and d
Ca 2p. Black solid line and red dashed line corresponded to the raw and fitted curve, respectively.

Figure 2.2 showed XPS spectra of AGC-PAA. The O 1s peak and Gd 4d peak were well shaped and
no obvious shift were observed. The two peaks at 142.4 eV and 150.4 eV were assigned to Gd 4d_{5/2} and
Gd 4d_{3/2} state, respectively. In light of the XPS results of AGC (Figure 2.3), the predominant peak at 531.9
10 eV corresponded to carbonate (-CO₃) oxygen atoms, which was in accordance with the one of ACC [Ref.

4. J. Am. Chem. Soc. 2018, 140, 14289-14299]. As compared to the Gd 4d spectrum of ACNC, the Gd
$4d_{3/2}$ peak (~ 150.1 eV) broadened and Gd $4d_{5/2}$ peak (~ 143.2 eV) shifted slightly.

**Figure 2.2 a** XPS survey spectrum of AGC-PAA, and the corresponding spectrum of **b** O 1s, **c** Gd 4d.

Black solid line and red dashed line corresponded to the raw and fitted curve, respectively.

**Figure 2.3 a** XPS survey spectrum of AGC, and the corresponding spectrum of **b** O 1s, **c** Gd 4d. Black
solid line and red dashed line corresponded to the raw and fitted curve, respectively.

**References for response letter:**

[Ref. 1.] Kiehl, J., et al. Grafting process of ethyltrimethoxysilane and polyphosphoric acid on calcium
carbonate surface. *Appl. Surf. Sci.* 264, 864-871 (2013).

[Ref. 2.] Alagumalai, K., et al. Impact of gadolinium oxide with functionalized carbon nanosphere: A
portable advanced electrocatalyst for pesticide detection in aqueous environmental samples. *Talanta* 238,
123028 (2022).

[Ref. 3.] Kiehl, J., et al. Efficient upconverting multiferroic core@shell photocatalysts: visible-to-near-
infrared photon harvesting. *ACS Appl. Mater. Interfaces* 9, 8142-8150 (2017).

[Ref. 4.] Du, H. C., et al. Amorphous CaCO₃: Influence of the formation time on its degree of hydration
and stability. *J. Am. Chem. Soc.* 140, 14289-14299 (2018).

2. The scale bars should be mentioned in the captions, such as Figure 1 d/e, Figure 6 c/e.

** Thank you for this valuable comment.

We have already added the information in the captions based on your suggestion.

3. The error bar in Figure 3e-3g & Figure 4c should be provided.

** Thank you for this valuable comment.

We have already supplied the information based on your suggestion, which improved the quality of our
paper. New Figure 3g (Figure 2.4) and 4c (Figure 2.5) were updated in the revised manuscript. Figure 3e
and 3f was not updated since they were single trial of ACNC's relaxivity under different magnetic field
strengths.

 **Figure 2.4** Quantitative analysis and comparison of the leakage of free gadolinium ion from ACNC and
 **PAA-Ca/Gd in NS** by means of ICP-AES.

 **Figure 2.5** Evolution of relative R_1 values from the zero time to each time ($R_1(t)/R_1(0)$) for three samples.

4. The semiquantitative analysis in Figure 5a&5c and Supplementary Figure S22&S23 should be provided.

**** Thank you for this valuable comment.**

In line with your suggestion, semiquantitative analysis was performed using ImageJ software. It is
clearly displayed that signal intensities in ACNC groups have a remarkable enhancement in comparison
to those in Gd-DTPA groups. The results of semiquantitative analysis for each graph were added to the
revised paper (Figure 2.6-2.11).

**Figure 2.6 a** Contrast enhanced MR angiography (MRA) images and **b** semiquantitative analysis of the
whole body on rat immediately after the bolus injection of (i) Gd-DTPA and (ii) ACNC (** $P < 0.001$).

**Figure 2.7 a** Contrast enhanced MR angiography (MRA) images and **b** semiquantitative analysis of the
whole body on rabbit immediately after the bolus injection of (i) Gd-DTPA and (ii) ACNC (** $P < 0.001$).

**Figure 2.8 a** Obliquesagittal (O) and Coronal (C) MRA images and **b** semiquantitative analysis of upper
body on beagle dog at immediately (IM) and 20 seconds after the bolus injection of (i) Gd-DTPA and (ii)
ACNC (** $P < 0.001$).

**Figure 2.9 a** MR angiography and **b** semiquantitative analysis on rat at different time points after the
intravenous injection of ACNC to analyse its dynamic distribution.

**Figure 2.10 a MR angiography and b semiquantitative analysis on rabbit at different time points after the**
**intravenous injection of ACNC to analyse its dynamic distribution (*P<0.05).**

**Figure 2.11 MR angiography and rapid renal clearance of ACNC. a MR angiography on beagle dog at**
**different time points after the intravenous injection of ACNC to analyse its dynamic distribution. The**
**kidney continues to brighten within 10 minutes and the bladder (the red arrow) brightened obviously after**

10 minutes. **b** selected areas within the yellow box and **c** semiquantitative analysis of selected areas to
analyse ACNC's dynamic distribution in kidneys and bladder region, respectively (*P<0.05).

5. The renal clearance of ACNC in mice should also be characterized.

** Thank you for this valuable comment.

To avoid the potential risk, renal clearance was the more desirable and preferred excretion route for

medical agents to minimal catabolism and body exposure [Ref. 5. ACS Nano 2015, 9, 6655-6674].

However, different with small molecular contrast agents, nanosized MR contrast agents, especially the

FDA approved iron oxide nanocrystals suffer from their poor renal excretion as the large size distribution

(>20 nm) [Ref. 6. Nat. Rev. Mater. 2018, 3, 358-374].

According to the recent review [Ref. 7. Chem. Soc. Rev. 2015, 44, 8576-8607], the size dependent

physiological barriers against nanoparticle blood circulation were schematically illustrated as shown in

Figure 2.12. In this figure, it is clearly described that liver and spleen are the major pathway for the

clearance of FDA approved nanosized iron oxide MR contrast agents including Feridex, Resovist and

ferumoxytol, because of their large hydrodynamic size between 21-160 nm. Furthermore, it is revealed

that NPs with dH < 10-15 nm could be eliminated via the kidney. In our previous research on ultrasmall

size of IONCs with the size about 10 nm, the renal clearance has been observed on both beagle dog and

monkey [Ref. 8. Nat. Biomed. Engineer. 2017, 1, 637-643].

**Figure 2.12** Schematic showing the size dependent physiological barriers against nanoparticles blood
 circulation. (from Ref. 7.: *Chem. Soc. Rev.* 2015, 44, 8576-8607).

In our research, MR angiography was used to characterize the renal clearance of ACNC. **Figure 2.13a**
 showed MR angiography on beagle dog at different time points after the intravenous injection of ACNC
 to analyse its dynamic distribution. The kidney continued to brighten within 10 minutes and the bladder
 (the red arrow) brightened obviously after 10 minutes. Notably, besides the conventional elimination from
 liver, the quick renal clearance of ACNC from blood vessels was observed in the MRA images after the
 intravenous injection. In addition, semiquantitative analysis was performed to analyse ACNC's dynamic
 distribution in kidneys and bladder region, respectively (**Figure 2.13b, c**). The significant boost of the
 signal in bladder region further validated the renal clearance of ACNC in beagle dog.

**Figure 2.13 MR angiography and rapid renal clearance of ACNC. a** MR angiography on beagle dog at
different time points after the intravenous injection of ACNC to analyse its dynamic distribution. The
kidney continues to brighten within 10 minutes and the bladder (the red arrow) brightened obviously after
10 minutes. **b** selected areas within the yellow box and **c** semiquantitative analysis of selected areas to
analyse ACNC's dynamic distribution in kidneys and bladder region, respectively (*P<0.05).

Herein, we conducted a pilot research on the clearance of our ACNC by MR imaging and analysis of
the urine from rat, because it is difficult to collect the whole urine from the large animal beagle dog or
small mouse. The urines were collected using metabolic cages (Figure 2.14a). In the urine collected 24
11 hours after the intravenous injection of ACNC, the content of gadolinium was detected by ICP-AES,
indicating the renal clearance efficiency of approximately 13% ID at 24h (Figure 2.14b). The renal
clearance efficiency at 24 h post injection in 5 rats was ~13.37%ID, ~11.69%ID, ~12.49%ID,
~13.84%ID, ~14.34%ID, respectively. Of course, more systematic researches are necessary to confirm
the renal clearance of ACNC in other animals, which is still ongoing.

**Figure 2.14 a** Metabolism cage to collect urine of rat. **b** Renal clearance efficiencies of ACNC in five rats
at 24h.

**References for response letter:**

[Ref. 5.] Yu, M. X. & Zheng, J. Clearance Pathways and Tumor Targeting of Imaging Nanoparticles. *ACS*
*Nano* 9, 6655-6674 (2015).

[Ref. 6.] Du, B. J., Yu, M. X. & Zheng, J. Transport and interactions of nanoparticles in the kidneys. *Nat.*
*Rev. Mater.* 3, 358-374 (2018).

[Ref. 7.] Arami, H., Khandhar, A., Liggitt, D. & Krishnan, K.M. In vivo delivery, pharmacokinetics,
biodistribution and toxicity of iron oxide nanoparticles. *Chem. Soc. Rev.* 44, 8576-8607 (2015).

[Ref. 8.] Lu, Y., et al. Iron oxide nanoclusters for T₁ magnetic resonance imaging of non-human primates.
*Nat. Biomed. Engineer.* 1, 637-643 (2017).

Reviewer #3 (Remarks to the Author):

The authors reported interesting finding for potential MRI agents, however, there are some important
issues, especially biological safety, I don't think this manuscript can be published.

Major issues:

1. Because of the amorphous state, the samples should be non-homogeneous microscopically. Therefore,
there is no evidence to show how much sample is actually involved in the tests, especially those involved
cells. High/low local concentrations of the samples could give a bias to the conclusion, and the
reproducibility of tissue/cell-related experiments might be poor.

** Thanks for your valuable comments.

The amorphous state refers to x-ray amorphous, which means that the material does not have a long
range order, which can be detected by X-rays. This does not mean that the samples are microscopically
non homogeneous. If we would have structurally non-homogeneous samples, we would have a broad
sedimentation coefficient distribution in Analytical Ultracentrifugation (AUC) since the sedimentation
coefficient depends on size and density, and the density is sensitive to the structure of the sample. They
all have a very similar size, density and thus structure since the distribution of the clusters is narrow. Since
AUC detects all sample particles, this result is statistically highly relevant. This is a proof that all cluster
samples are the same on time scales of hours and therefore all clusters are involved in the tests. However,
to understand the homogeneity of the cluster samples, it needs to be understood that they are basically
better expressed as highly dynamic polymers, which continuously form and break the Calcium-carbonate
bonds. This dynamic situation is described in one paper [Ref. 1. Nat. Commun. 2011, 2, 590]. However,

these fluctuations are on the nm time scale and average out for time scales on which laboratory work is
done so that they are homogenous samples in the analytics, which are the time average structures of these
very fast fluctuations. Our experimental results have demonstrated excellent reproducibility.

References for response letter:

[Ref. 1.] Demichelis, R., Raiteri, P., Gale, J. D., Quigley, D. & Gebauer, D. Stable prenucleation mineral
clusters are liquid-like ionic polymers. *Nat. Commun.* 2, 590 (2011).

2. Not like other nanoclusters applied as MRI contrast agents, there is no indication of “relaxivity density”
(defined as [relaxivity per unit]/[unit volume or molecular weight]) or “relaxivity per Gd” (commonly
used to judge Gd-chelates). Those parameters are more specific and more applicable than an overall
relaxivity value, especially for the macromolecules studied here.

** Thank you for this valuable comment.

Based on your suggestion, we have reviewed the relevant literature on "relaxivity density". According
to the literatures [Ref. 2. *Curr. Top. Med. Chem.* 2013, 13, 411-421; Ref. 3. *J. Med. Chem.* 2011, 54,
5185-5194], "relaxivity density" was define as (relaxivity per nanoparticle)/(nanoparticle volume or
molecular weight) (Figure 3.1).

Table 1. Physical and Magnetic Properties of Selected Gd-based Nanoparticles and Macromolecules

Nanoparticle	Hydrodynamic Diameter (nm)/MW(KDa)	#Gd/particle	r_1 /Gd ($\text{mM}^{-1}\text{s}^{-1}$)	r_1 /particle ($\text{mM}^{-1}\text{s}^{-1}$)	Relaxivity Density	Ref.
Poly(L-lysine)-gadotetate dimeglumine	480 KDa	40	10.8 (0.4T, 39 °C)	432	0.9 ($\text{mM}^{-1}\text{s}^{-1}/\text{KDa}$)	[31]
Gd-DTPA-dextran	75 KDa	187	10.5 (0.25T, 37°C)	1,964	26.2 ($\text{mM}^{-1}\text{s}^{-1}/\text{KDa}$)	[36]
Cascade-Gd-DTPA-24	30 KDa	24	10 (2T, 37 °C)	240	8 ($\text{mM}^{-1}\text{s}^{-1}/\text{KDa}$)	[42, 46]
Gd-polydisulfide copolymer (cystine) GDCP	22 KDa	35	6.8 (3.0T)	238	10.8 ($\text{mM}^{-1}\text{s}^{-1}/\text{KDa}$)	[47-51]
G5-PAMAM dendrimer	5.4 nm/118 KDa	96	30 (20 MHz, 23°C)	2,880	34.9 ($\text{mM}^{-1}\text{s}^{-1}/\text{nm}^3$) 24.4 ($\text{mM}^{-1}\text{s}^{-1}/\text{KDa}$)	[53]
G10-PAMAM dendrimer	13.5 nm/3000 KDa	1860	36 (20 MHz, 23°C)	66,960	51.8 ($\text{mM}^{-1}\text{s}^{-1}/\text{nm}^3$) 22.3 ($\text{mM}^{-1}\text{s}^{-1}/\text{KDa}$)	[53]
Dendrimer nanocluster (DNC)	150 nm	300,000	12.3 (1.41T, 40°C)	3,600,000	2.0 ($\text{mM}^{-1}\text{s}^{-1}/\text{nm}^3$)	[58, 59]
Paramagnetic porous polymersome	130 nm	40,000	7.5 (1.41T, 40°C)	300,000	0.26 ($\text{mM}^{-1}\text{s}^{-1}/\text{nm}^3$)	[67, 68]
Micelle (shell-crosslinked knedel-like (SCK) nanoparticle	40±3 nm	513	39 (0.47T, 40°C)	20,000	0.6 ($\text{mM}^{-1}\text{s}^{-1}/\text{nm}^3$)	[74]
Perfluorocarbon (PFC) emulsion Gd-MeO-DOTA-PE	190 nm	49,329	29.8 (0.47T, 40°C)	1,470,000	0.4 ($\text{mM}^{-1}\text{s}^{-1}/\text{nm}^3$)	[78]
PEG-US-Gd ₂ O ₃	2.8±1.1 nm	200	9.4 (1.5T, 20°C)	1880	163.6 ($\text{mM}^{-1}\text{s}^{-1}/\text{nm}^3$)	[87]
Gadonanotube	20-80 nm	100	180 (1.5T, 37°C)	18000	0.21 ($\text{mM}^{-1}\text{s}^{-1}/\text{nm}^3$)	[91]
Virus-like particle (VLP) P22 _{S90C} -xAEMA-Gd	71±3 nm	9,100±800	22 (1.4T)	200,000	1.07 ($\text{mM}^{-1}\text{s}^{-1}/\text{nm}^3$)	[95, 96]
Low density lipoprotein (LDL) nanoparticle	26.3 nm	180	8.1±0.19 (60MHz, 40°C)	1440	0.15 ($\text{mM}^{-1}\text{s}^{-1}/\text{nm}^3$)	[99]

Figure 3.1 A summary of the physical and magnetic properties of various macromolecule and
nanoparticle-based MR contrast agents reported in one review. (from Ref. 2.: Curr. Top. Med. Chem. 2013,
13, 411-421).

In our previous studies, we have obtained the molecular weight of ACNC by the use of AUC
measurements (Table 1). We dissolved a weighted amount of ACNC lyophilized powder with freshly
prepared chloroazotic acid ($\text{HNO}_3/\text{HCl} = 3:1$) and brought it to a certain volume of aqueous solution. The

value of Gd/Nanoparticle can be thus calculated by measuring the amount of Gd in the known mass of the
 sample via ICP-AES.

We calculated the value of r_1/Gd in our previous tests. The numerator of the equation ($r_1/\text{Nanoparticle}$)
 can be calculated by multiplying the known product of r_1/Gd and Gd/Nanoparticle. AUC experiment
 provided the diameter and molecular weight of ACNC. The molar mass of the nanoparticles with a
 diameter of 1.5 nm is 2230 g/mol. " Relaxivity density " of ACNC can be eventually calculated with the
 help of the volume and molecular weight ($21.46 \text{ mM}^{-1}\text{s}^{-1}/\text{nm}^3$, $17.01 \text{ mM}^{-1}\text{s}^{-1}/\text{KDa}$, respectively). Those
 values calculated in each step were recorded in Table 2. Compared with some Gd-based nanoparticles and
 macromolecules that with relaxivity density already known, ACNC performed excellent in terms of
 relaxivity density.

Sample	s (10^{-13}s)	D 10^{-10} (m^2/s)	d_H (nm)	M (g/mol)	ρ (g/cm^3)	\bar{v} (cm^3/g)	f/f ₀	δ ($\text{gH}_2\text{O}/\text{gSolute}$)
ACNC1	1.19	2.86	1.5	2030	2.0171	0.4958	1.90	2.90
ACNC2	1.20	2.80	1.5	2200	2.0045	0.4989	2.00	3.49
ACNC3	1.17	2.60	1.6	2460	1.8157	0.5508	1.90	3.22
Average	1.19	2.75	1.5	2230	1.9458	0.5152	1.93	3.20

Table 1 Results of AUC experiments. Results of AUC experiments using 2-dimensional spectrum analysis
 (2-DSA) for the determination of the cluster hydration calculated for spherical clusters via the Perrin
 coefficient. Absorption optics was used for ACNC detection via PAA (230 nm). s = sedimentation
 coefficient, D = diffusion coefficient, ρ = density of the cluster (n=3) and δ = hydration.

Sample	Hydrodynamic Diameter (nm)	MW(KDa)	#Gd/particle	r_1/Gd ($\text{mM}^{-1}\text{s}^{-1}$)	$r_1/\text{particle}$ ($\text{mM}^{-1}\text{s}^{-1}$)	Relaxivity Density ($\text{mM}^{-1}\text{s}^{-1}/\text{nm}^3$)	Relaxivity Density ($\text{mM}^{-1}\text{s}^{-1}/\text{KDa}$)
ACNC1	1.5	2.03	1	38.38	38.38		
ACNC2	1.5	2.20	1	38.19	38.19		
ACNC3	1.6	2.46	1	37.21	37.21		
Average	1.5	2.23	1	37.93	37.93	21.46	17.01

Table 2 Physical and Magnetic Properties of ACNC. "Relaxivity Density" of ACNC can be eventually
calculated with the help of the volume and molecular weight

References for response letter:

[Ref. 2.] Huang, C. H. & Tsourkas, A. Gd-based macromolecules and nanoparticles as magnetic resonance
contrast agents for molecular imaging. *Curr. Top. Med. Chem.* 13, 411-421 (2013).

[Ref. 3.] Řehoř, I., et al. Phosphonate-titanium dioxide assemblies: platform for multimodal diagnostic-
therapeutic nanoprobes. *J. Med. Chem.* 54, 5185–5194 (2011).

3. Because the structure of ACNC is not uniform and not clearly reported, it might be difficult to propose
further modifications on this material. Not like nanoparticles or complexes which can be coated or
conjugated with vectors for targeted therapy/imaging, there seems little room for the amorphous materials.

** Thank you for this valuable comment.

As mentioned in Question 1, the time dependent structure of ACNC has been reported on basis of
simulations, which are able to characterize the fast structural fluctuations, demonstrating homogeneous
and uniform structure of ACNC. ACC clusters have also been successfully coated with silica to fix their
structures [Ref. 4. *Adv. Funct. Mater.* 2012, 22, 4301]. Afterwards their size could be investigated as
defined by scattering techniques as well as the size of cluster aggregates. There is no reason to believe that
addition of gadolinium to the calcium carbonate clusters would change this situation and this is supported
by the similar results from AUC for the clusters with or without Gadolinium. For the data of pure CaCO_3
clusters, we can see the previous report [Ref. 5. *Science* 2008, 322, 1819]. They are essentially the same
as the here reported AUC data for the clusters with Gadolinium. Therefore, the clusters could easily be
coated with silica as reported in one paper [Ref. 4. *Adv. Funct. Mater.* 2012, 22, 4301] and any other
molecule can be bound to their surface via established silane chemistry including their conjugation with
vectors. Therefore, the ACNCs are a very promising species for further applications after performing the
literature reported silica coating. In addition, doxorubicin hydrochloride (DOX) and chlorin e6 (Ce6) could
be condensed into amorphous calcium carbonate (ACC) via calcium ion-mediated coordination.
Establishment of this facile method not only optimizes delivery of multi-therapeutics, but also elicits
enhanced chemo-/photodynamic tumor therapy, which has been reported by our group recently [Ref. 6.
*Nano Today*, 2022, 43, 101382].

Herein, fluorescein isothiocyanate-labeled BSA (BSA-FITC) was used as the model protein to
evaluate whether ACNC could be used to conjugate protein. Briefly speaking, according to previous
literatures, ACNC were bonded with BSA-FITC using EDC/NHS covalent chemistry. The products were
purified via centrifugation (Figure 3.2a, b). Fluorescence spectra of BSA-FITC, ACNC, and ACNC-BSA-
FITC were detected to characterize whether BSA-FITC could be modified on the surface of ACNC. As

shown in Figure 3.2c, BSA-FITC has a characteristic absorption at ~550 nm, and appearance of the
characteristic peak of ACNC-BSA-FITC confirmed the success of modification.

Then, we attempted to load drug on ACNC using doxorubicin hydrochloride (DOX). ACNC-DOX
was prepared by simply mixing ACNC with Dox, purifying by centrifugation, and washing with deionized
water (Figure 3.2d, e). Figure 3.2f shows UV-Visible absorbance spectra of DOX, ACNC and ACNC-
DOX. DOX presented a characteristic absorption peak at around 480 nm, and the peak was retained after
DOX being loaded onto ACNC. Overall, the successful modification of protein and drug serves as strong
supporting evidence indicating that ACNC could be a promising type of nanoagent with great potential
for theranostics.

**Figure 3.2** Photographs of BSA-FITC, ACNC, and ACNC-BSA-FITC under **a** room lighting and **b** UV
illumination; **c** Fluorescence spectra of BSA-FITC, ACNC, and ACNC-BSA-FITC dispersed in aqueous
solution; Photographs of DOX, ACNC and ACNC-DOX under **d** room lighting and **e** UV illumination; **f**
UV -Vis spectra of DOX, ACNC and ACNC-DOX dispersed in aqueous solution.

References for response letter:

[Ref. 4.] Kellermeier, M., et al. Colloidal stabilization of calcium carbonate prenucleation clusters with
silica. *Adv. Funct. Mater.* 22, 4301-4311 (2012).

[Ref. 5.] Gebauer, D., Volkel, A. & Cölfen, H. Stable prenucleation calcium carbonate clusters. *Science*
322, 1819-1822 (2008).

[Ref. 6.] Yan, B. B., et al. All-in-one hollow nanoformulations enabled imaging-guided Mn-amplified
chemophototherapy against hepatocellular carcinoma. *Nano Today* 43, 101382 (2022).

4. The stability tests are not comprehensive. It is not surprising that ACNC did not leak free Gd(III) ions
obviously in aqueous solution or in in vitro tests. However, there at least should be a competition test
between ACNC and other Gd(III) chelators. For examples, the competition between a new Gd(III)
complex and excessive DTPA free ligand is a classical way to compare the stability.

** Thank you for this valuable comment.

We addressed this issue by applying DTPA in the ligand competition assay. 1 mL ACNC aqueous
solution (5 mM Gd) was added into 1 mL DTPA solution (5 mM) and the homogeneous solution was taken
for measurement. Compared with the commercial Gd-DTPA, the concentration of Gd-DTPA solutions was
identical to ACNC solutions as 5 mM Gd.

In addition, to assess the relaxivity stability of ACNC, the longitudinal relaxation time was monitored.
There was no observable evidence showing that relaxivity of ACNC was significantly altered, which
confirmed that ACNC was not affected by the competition between Gd(III) complex and excessive DTPA
free ligand (Figure 3.3).

 **Figure 3.3** Evolution of relative R_1 values in a competition test from the zero time to each time ($R_1(t)/R_1(0)$)
 for ACNC and Gd-DTPA. DTPA served as the competing ligand.

 Besides, although Fig. 3G is described as results for stability tests (line 165). Its picture caption and axis
 labels are unclear and there is no information on how this experiment was conducted.

**** Thank you for this valuable comment.**

Our previous presentation may result in unclearness due to description missing. The clarification and
 other relevant information have been added in order to help readers facilitate the process of understanding
 our experimental approach.

As we known, the gadolinium linear chelates such as Gd-DTPA and Gd-DTPA-BMA possess poor
 stability. Due to abundant endogenous ions in bodies, high kinetic inertness against transmetallation is
 critically required for intravenously injectable agents. Among these endogenous ions (Cu^{2+} , Ca^{2+} , Fe^{3+} and
 Zn^{2+}), Zn^{2+} exhibited both relatively high concentration (55-125 mM) in blood and comparable association

constant towards ligands ($L_{(n)}^-$, such as DTPA, DOTA) to gadolinium [Ref. 7. *Anal. Chem.* 2008, 80, 8163-
8170]. In the presence of Zn^{2+} , it would compete with gadolinium to chelate with ligands, leading to the
leakage of free gadolinium ion. Then leaked gadolinium ions will be captured by phosphate anions in PBS
buffer to form insoluble gadolinium phosphate, resulting in the obvious increase the T_1 relaxation time of
the dispersion. We learn from the systematic research on stability of gadolinium based contrast agents by
Laurent and Muller [Ref. 8. *Invest. Radiol.* 2001, 36, 115-122; Ref. 9. *Contrast Media Mol. Imaging*
2006, 1, 128-137]. Then we evaluated the *in vitro* transmetallation of ACNC and Gd-DTPA following the
standard procedure by Laurent and Muller [Ref. 10. *Contrast Media Mol. Imaging* 2010, 5, 305-308].

The experiments were performed as follows: ACNC, Gd-DTPA and PAA-Ca/Gd samples were
freshly prepared. At $t=0$, each sample (2.5 mM Gd) and $ZnCl_2$ aqueous solution (250 mM, 20 μ L) were
mixed in a 2 mL phosphate buffer. Then, 1 mL mixed solution was contained in a chromatographic bottle
for measurement. The measurements were carried out at 37 °C under a 0.5 T NMI20-Analyst NMR
Analyzing and Imaging system. TR/TE = 100/5.6 ms. Longitudinal relaxation times were measured at
different points in time. The relaxation rate at $t=0$ (denoted as $R_1(0)$) was calculated by the formula
$R_1=(1/T_1)$. The relaxation rates at other time points were all respectively calculated and recorded as
corresponding $R_1(t)$, for the purpose of assessing the 2-day transmetallation through monitoring the ratio
of $R_1(t)/R_1(0)$.

As a result, linear gadolinium chelates such as Gd-DTPA exhibited a sharp decrease of the measured
relaxation rate (R_1) in this physiological environment-mimic Zn^{2+} involved PBS buffer. Compared with
Gd-DTPA, ACNC exhibited obvious enhancement of the stability against the zinc mediated
transmetallation effect, indicating the lower dissociation risk for ACNC in the retention period *in vivo*. For
contrast, a chelate composed of PAA, calcium and gadolinium ion (namely PAA-Ca/Gd) was synthesized

to investigate the contribution of ACC like environment to the physiological stability. Notably, this PAA-
 Ca/Gd chelate in the absence of ACC like environment exhibited even faster leakage than Gd-DTPA
 (Figure 3.4). After exposure to Zn^{2+} for 48h, the relative R_1 value ($R_1(t)/R_1(0)$) was decreased to about
 60%, indicating gadolinium chelated with PAA could be displaced by zinc ion easily. The weak inertness
 of PAA-Ca/Gd chelates without the formation of ACC environment further confirmed the excellent
 inertness of ACNC should be primarily attributed to the confinement of gadolinium in ACC, but not the
 introduction of polyacrylic acid.

 Figure 3.4 a The possible transmetalation of a Gd complex by diamagnetic Zn^{2+} ions occurs in such a
 PBS buffer solution (from Ref. 10.: Contrast Media Mol. Imaging 2010, 5, 305-308). b Evolution of
 relative R_1 values from the zero time to each time ($R_1(t)/R_1(0)$) for three samples.

 References for response letter:
 [Ref. 7.] Kunnemeyer, J., et al. Speciation analysis of gadolinium-based MRI contrast agents in blood
 plasma by hydrophilic interaction chromatography/electrospray mass spectrometry. *Anal. Chem.* 80, 8163-
 8170 (2008).

[Ref. 8.] Laurent, S., Elst, L.V., Copoix, F. & Muller, R.N. Stability of MRI paramagnetic contrast media
- a proton relaxometric protocol for transmetallation assessment. *Invest. Radiol.* 36, 115-122 (2001).

[Ref. 9.] Laurent, S., Elst, L.V. & Muller, R.N. Comparative study of the physicochemical properties of
six clinical low molecular weight gadolinium contrast agents. *Contrast Media Mol. Imaging* 1, 128-137
(2006).

[Ref. 10.] Laurent, S., Vander Elst, L., Henoumont, C. & Muller, R.N. How to measure the transmetallation
of a gadolinium complex. *Contrast Media Mol. Imaging* 5, 305-308 (2010).

5. It could be difficult to precisely control the administration amount in practical uses (minimal dosage)
because the Gd content in a tiny amount of amorphous materials could vary largely and then the
performance will be quite different.

** Thank you for this valuable comment.

In practice, the amount of Gd ions was quantitatively determined by inductive coupled plasma atomic
emission spectrometer (ICP-AES). The detection limit of ICP-AES was 0.001 µg/mL. ACNC acts
essentially on the application of this method as a contributor by delivering favorable dispersity and stability.
The commercial Gd-DTPA used as a control subject can also be precisely measured by this method. To
verify the validity and reproducibility of the method, a batch of half-diluted gradient samples were
fabricated and tested by ICP-AES. The results showed that the ACNC samples and commercial products
at all concentration levels have demonstrated excellent accuracy with very small errors (Figure 3.5).
Therefore, we believe that our samples can be formulated with precise concentrations for further
administration in practical applications.

 **Figure 3.5** Concentration validation experiments. Concentrations of a batch of half-diluted gradient
 **samples (ACNC and Gd-DTPA, n=3) measured by ICP-AES.**

 6. Mentioned in the manuscript is that most ACC are instable in aqueous solution which limits their
 applications, but the authors report ACNC to be biocompatible. It is necessary and will be impactful to
 explain in detail how the added Gd(III) could dramatically change the physical properties.

**** Thank you for this valuable comment.**

**This suggestion is very useful for us to investigate the stability of Gd occluded ACC complex.**
 **According to this suggestion, the transformations of ACC-Gd were investigated in different mediums**
 **(ethanol, water) to further explore how Gd affected the stability of ACC (Figure 3.6).**

**Figure 3.6** XRD patterns of ACC and ACC-Gd obtained from different stages in ethanol, water. The signs
represent the characteristic peak of diffraction patterns from the PDF card (vaterite (Δ , PDF: 33-0268);
calcite (*, PDF: 05-0586); Aragonite (\bullet , PDF: 24-0025); gadolinium carbonate hydrate (\square , PDF: 37-
0559)).

First of all, the aqueous suspensions were centrifuged at 3000 rpm for 2 min immediately after the
addition of ethanol in our experiment. Then, stabilities of both additive-free ACC and ACC-Gd were
studied in ethanol and water, respectively. The XRD results of samples were collected at different time
points.

(1) After precipitates were further washed twice by ethanol, both products were dispersed in ethanol. ACC
would formed multiple phases eventually and vaterite was involved in this process as an intermediate

phase. After 6 months, calcite, aragonite and vaterite were all contained in the product. However, ACC-
Gd was stable in ethanol over 6 months.

(2) After precipitates were further washed twice by water, both products were dispersed in water. ACC
transformed to calcite within one day. As a thermodynamically stable crystalline form, the calcite phase
would be maintained over the course of the next month. In comparison, the fresh-prepared ACC-Gd could
maintain amorphous phase for tens of minutes. Then, a characteristic peak of calcite could be observed
within 1 hour. 30 days later, a calcite phase and a gadolinium carbonate hydrate phase were eventually
formed [Ref. 11. Mater. Chem. Phys. 2007, 106, 149-157].

Overall, Gd should play a reasonable role in retarding the crystallization of ACC.

Although a crystalline phase was observed in ACC-Gd dispersed in water, there was still higher water
content was held. However, in comparison with ACC-Gd, the water in ACC was rarely held. According to
the TGA result, the weight loss before 300 °C could be calculated as water content (Figure 3.7). The
weight loss of ACC-Gd and ACC powder exposed to dry air over 6 months was more than 10% and only
0.1%, respectively. Note that ACC, which was precipitated in a hydrated form has completely lost its
hydration water under these conditions.

**Figure 3.7** Thermogravimetric curve of ACC and ACC-Gd lyophilized powder isolated by deionized
water and then exposed to dry air for over 6 months. The weight loss before 300 °C could be attributed to
the loss of water.

In the isolation process of ACC composites, both additive-free ACC and ACC-Gd were washed by
ethanol. The changes of morphology of ACC and ACC-Gd after dispersed in ethanol in two weeks were
observed by TEM (Figure 3.8). A disordered emulsion-like structure was formed initially in additive-free
ACC, as well as in ACC-Gd with smaller particle size. For ACC, branched aggregates with large particles
were occurred 3 days later and several micrometers sized crystals could be observed one week later,
indicating the transformation from small ACC nanoparticles into thermodynamically stable form and
further ripening. Two weeks later, few small nanoparticles were retained and larger crystals were formed.
In contrast, no obvious growth and aggregate of particle was observed in ACC-Gd.

**Figure 3.8** TEM images of fresh prepared ACC and ACC-Gd dispersed in ethanol, and images at different
time points after dispersed in ethanol of 3 day, 1 week, and 2 week, respectively

The digital photos of ACC and ACC-Gd samples dispersed in ethanol and aqueous solution were
showed as follows respectively (Figure 3.9). Both ACC and ACC-Gd could dispersed well in ethanol.
However, compared with our stable ACC-Gd, there is obvious sediment at the bottom of bottle for ACC
dispersion in ethanol over 15 days. Moreover, there is obvious difference between ACC and ACC-Gd after
dispersed in water. ACC would precipitate rapidly at the bottom of bottle in 5 minutes if it was dispersed
in water, and a calcite phase could be detected by XRD analysis (Figure 3.6). However, when ACC-Gd
was dispersed into aqueous solution, it would disperse well at first. Then, an aggregation of ACC-Gd could
be observed in the next 2 hours to form a gel-like flocculation without any sediment. The changes of
morphology of ACC and ACC-Gd dispersed in water for a long time were also observed by TEM (Figure
3.10). In the TEM images, few emulsion-like ACC could be found in ACC dispersed in water, and micro-
sized calcite formed quickly in the next few days, while ACC-Gd aqueous dispersion could maintain the
emulsion-like morphology for quite a long time.

**Figure 3.9** The digital photos of ACC and ACC-Gd samples dispersed in **a** ethanol and **b** water. The red
arrow in **a** showed the sediment of ACC dispersed in ethanol over 15 days.

**Figure 3.10** TEM images of fresh ACC and ACC-Gd dispersed in water, and images at different time
points after dispersed in ethanol of 3 day, 1 week, and 2 week, respectively.

In the typical synthesis process of ACC-Gd, gadolinium chloride hexahydrate was mixed with the
calcium chloride solution at the $[Ca^{2+}]:[Gd^{3+}]$ molar ratio of 5:1. Gd content in ACC-Gd was measured by
ICP-AES, the accurate mass fraction of Gd occluded in ACC-Gd was measured to be about 25%, and the
mole ratio of $[Ca]:[Gd]$ in as-prepared ACC-Gd was about 4:1 calculated by ICP-AES result. FT-IR
spectrum showed a consistency compared with the amorphous calcium carbonate phase, and the ν_2
vibration shifted to lower value indicated the formation of amorphous gadolinium carbonate (AGC) phase
(Figure 3.11a) [Ref. 11. Mater. Chem. Phys. 2007, 106, 149-157]. The ^{13}C nuclear magnetic resonance
(NMR) result of ACC-Gd showed a characteristic vibration at 168 ppm, which indicated the ACC
precursor phase in ACC-Gd (Figure 3.11b).

Furthermore, when we studied the short-range calcium environments of ACNC composite by X-ray
Absorption Spectroscopy (XAS) measurement and extended X-ray absorption fine structure (EXAFS)
analysis, we also study suitable control sample containing ACC including ACC-PAA, ACC-Gd and
another control sample not containing ACC (namely PAA-Ca). In addition, the control gadolinium sample
not containing ACC (namely PAA-Ca/Gd) was analysed in order to identify the impact of the presence of
gadolinium ions on the short-range structure of the precipitated complex without ACC. We performed
XAS and EXAFS on the XAFS beamline at Elettra Synchrotron (Trieste, Italy) with the assistance of Prof.
Denis Gebauer, an expert of ACC formation in Leibniz Universität Hannover, Germany. XAS revealed
that the calcium environments within the ACC standard, ACC-Gd and ACNC samples closely relate to
those in previously investigated ACCs [Ref. 12. *Angew. Chem. Int. Ed.* 2016, 55, 8117-8120; Ref. 13.
*Angew. Chem. Int. Ed.* 2010, 49, 8889-8891; Ref. 14. *Chem. Mater.* 2008, 20, 4720-4728; Ref. 15. *Angew.*
*Chem. Int. Ed.* 2016, 55, 12206-12209], while PAA-Ca and PAA-Ca/Gd presented significantly different
calcium coordination environments. Moreover, the evaluations showed that the Ca-O₁ short-range
environments in ACC-Gd and ACNC were essentially identical with that in the ACC-PAA standard,
strongly suggesting that these samples contain distinct, ACC-like domains. Again, these are structurally
similar to ACC-Gd and also other, additive-free ACCs reported in the literature. Finally, the incorporation
of gadolinium ions within synthetic ACC and ACC-PAA did not seem to significantly impact the short
range order of these samples (Figure 3.11c, d).

Based on the XRD, FT-IR, NMR and EXAFS data, the composition of ACC-Gd includes amorphous
calcium carbonate and amorphous gadolinium carbonate, which should form a hydrated amorphous
precursor phase similar to Mg-ACC widespread in nature.

 **Figure 3.11** a FT-IR spectra of amorphous gadolinium carbonate and ACC-Gd. b ^{13}C solid-state NMR
 spectra of ACC-Gd and ACC recorded by hpdec spectra. c k^2 -weighted EXAFS and d k^2 -weighted Fourier
 transform of the EXAFS for the ACC standard, ACC-Gd and ACNC, ACC-PAA, PAA-Ca and PAA-Ca/Gd.
 Black lines are experimental data and dotted red lines their best fit.

 **References for response letter:**
 [Ref. 11.] Park, I. Y., Kim, D., Lee, J., Lee, A. H. & Kim, K. J. Effects of urea concentration and reaction
 temperature on morphology of gadolinium compounds prepared by homogeneous precipitation. *Mater.*
 *Chem. Phys.* 106, 149-157 (2007).
 [Ref. 12.] Farhadi-Khouzani, M., Chevrier, D. M., Zhang, P., Hedin, N. & Gebauer, D. Water as the key
 to proto-aragonite amorphous CaCO_3 . *Angew. Chem. Int. Ed.* 55, 8117-8120 (2016).

[Ref. 13.] Gebauer, D., et al. Proto-calcite and proto-vaterite in amorphous calcium carbonates. *Angew.*
*Chem. Int. Ed.* 49, 8889-8891 (2010).

[Ref. 14.] Michel, F. M., et al. Structural characteristics of synthetic amorphous calcium carbonate. *Chem.*
*Mater.* 20, 4720-4728 (2008)

[Ref. 15.] Sun, S. T., Chevrier, D. M., Zhang, P., Gebauer, D. & Colfen, H. Distinct short-range order is
inherent to small amorphous calcium carbonate clusters (< 2 nm). *Angew. Chem. Int. Ed.* 55, 12206-12209
(2016).

7. Other cell lines for MTT may be expected. Only one cell line for biocompatibility seems not convincing
enough. In addition, different incubation times may also be required in this experiment.

** Thank you for this valuable comment.

On the basis of your suggestion, extra cytotoxicity experiments were conducted on human renal
tubular epithelial cell (HK-2) and human immortal keratinocyte (HaCaT) lines. We also extended
incubation time to 48h to determine cell viabilities of HK-2 and HaCaT. The fact of ACNC exhibited cell
viability >97% against all types of cells at concentrations up to 500 µg (Gd)/mL indicated ACNC's
capability of cytocompatibility (Figure 3.12).

**Figure 3.12** MTT results of ACNC evaluated on HK-2 and HaCaT cells at a 24 h and b 48 h.

8. For the confocal imaging (Fig. S14 and Fig. S15, p10, line190-192), Why are the incubation times for
these two experiments different? In addition, results with longer incubation time will be more convincing
for this study.

**** Thank you for this valuable comment.**

**In line with your suggestion, HK-2 cells were cultured for 48 hours to assess the cytotoxicity including**
**additional markers (mitochondrial, nuclear DNA and cellular proliferation). Gd-DTPA with identical**
**concentrations served as control.**

**To determine whether damage of nuclear DNA occurred, we used TdT-mediated dUTP Nick-End**
**Labeling (TUNEL) method to examine apoptotic cells. Apoptotic cells were labeled by red fluorescent**
**probe Cyanine 3 (Cy3). The results from the immunofluorescent TUNEL staining assay of HK-2 (Figure**
**3.13) showed that ACNC did not cause DNA damage even at a high concentration of 500 µg (Gd)/mL.**

**Figure 3.13** Representative TUNEL immunofluorescent images of HK-2 cells separately incubated with
ACNC or Gd-DTPA for **a** 24 h and **b** 48 h, scale bar = 50 μ m.

An EdU (5-Ethynyl-2'-deoxyuridine) assay was performed to explore the impact of ACNC on
cellular proliferation. Logarithmic growth stage HK-2 cells with corresponding concentration of EdU
reagent and ACNC were seeded in 6-well plate. As shown in Figure 3.14, like Gd-DTPA, addition of
ACNC has very little effects on cell proliferation.

**Figure 3.14** Cell proliferation in HK-2 cells by EdU cell proliferation assay, scale bar = 50 µm. The cells
were incubated with ACNC or Gd-DTPA for **a** 24 h and **b** 48 h, respectively.

**Altered mitochondrial homeostasis is considered an early event that precedes activation of apoptosis.**
**It is more meaningful to determine the mitochondrial membrane potential at earlier time points [Ref. 16.**
**Dev. Cell 2001, 1, 515-525; Ref. 17. Cell 2006, 125, 1241-1252]. Therefore, the incubation time of cell**
**proliferation assay (12 h) was shorter than that of cell cytotoxicity and proliferation assay (24 h) in our**
**previous studies.**

In addition, one shorter incubation time point (6 h) was added to assess mitochondria damage of cells
[Ref. 18. Proc. Natl. Acad. Sci. U. S. A. 1991, 88, 3671-3675; Ref. 19. Sci. Adv. 2010, 6, eaay7608]. We
applied J-aggregates (JC-1) using mitochondrial membrane potential assay kit with JC-1 to investigate the
mitochondrial membrane potentials (MMP). When the mitochondrial membrane potential decreased, JC-
1 in a monomeric form emitted green fluorescence, which was different from the red fluorescence emitted
by the aggregates in the normal mitochondria. As shown in **Figure 3.15**, strong red fluorescence and weak
green fluorescence were observed after separate treatments with ACNC and Gd-DTPA at varying
gadolinium concentrations for 6 h, demonstrating healthy mitochondria of HK-2 cells at the earlier time
points.

 **Figure 3.15** CLSM images of HK-2 cells stained with JC-1 after treatment with ACNC or Gd-DTPA for
 **a 6 h and b 12 h, respectively, scale bar = 50 μ m.**

 **References for response letter:**

**[Ref. 16.] Frank, S., et al. The role of dynamin-related protein 1, a mediator of mitochondrial fission, in**
 **apoptosis. *Dev. Cell* 1, 515-525 (2001).**

**[Ref. 17.] Chan, D. C., et al. Mitochondria: Dynamic organelles in disease, aging, and development. *Cell***
 **125, 1241-1252 (2006).**

[Ref. 18.] Smiley, S. T., et al. Intracellular heterogeneity in mitochondrial-membrane potentials revealed
by a J-aggregate-forming lipophilic cation JC-1. *Proc. Natl. Acad. Sci. U. S. A.* **88**, 3671-3675 (1991).

[Ref. 19.] Michel, F. M., et al. Bioenergetic-active materials enhance tissue regeneration by modulating
cellular metabolic state. *Sci. Adv.* **6**, eaay7608 (2020)

9. For the clearance study, in Fig.S21, the amounts of ACNC remaining in the liver and spleen in mice
after even 30 days were not negligible and could be harmful.

** Thank you for this valuable comment.

Previous studies reported that the majority of intravenously administered nanoparticles is removed
from the bloodstream by cells of the mononuclear phagocyte system (MPS), which is defined as a network
of immune and architectural cells and found to be located in organs such as liver, spleen, etc [Ref. 20. *Nat.*
*Rev. Mater.* 2016, 1, 16014]. The majority of administered nanomaterials were sequestered by liver and
spleen, which thus become the major biological barriers for translating nanomedicines [Ref. 21. *Nat.*
*Mater.* 2016, 15, 1212-1221].

2 h after injection, the amount of nanoparticles excreted into the urine would experience
exponentially reduction, and consequently accumulate in organs such as liver and spleen because of the
increasing nanoparticle core density in the early renal elimination stage [Ref. 22. *Angew. Chem. Int. Ed.*
2016, 55, 16039-16043; Ref. 23. *Nat. Rev. Mater.* 2018, 3, 358-374]. There are also limitations that are
worth noticing. The nanomaterials released from other organs or tissues would return to the liver and be
captured again. Meanwhile, the accumulation of administered nanomaterials in other organs or tissue
(spleen, tumor, etc.) could be caused by the release of NMs moving from liver sinusoid to the systemic

circulation via the central vein [Ref. 21. *Nat. Mater.* 2016, 15, 1212-1221; Ref. 24. *J. Controlled Release*
2016, 240, 332-348]. Macrophages in the liver and spleen are able to internalize nanomaterials and hold
them for a longer period of time in the body. Due to the reduced velocity of nanomaterials (1,000-fold) in
the liver sinusoids, their retention time is further enhanced [Ref. 21. *Nat. Mater.* 2016, 15, 1212-1221].

Nanomaterials detained in the liver could be eliminated from the liver via hepatobiliary clearance
[Ref. 25. *Adv. Mater.* 2022, 34, 2106456]. Nanomaterials in bulk could be excreted by hepatocytes through
emptying of lysosomal contents into the bile [Ref. 26. *Hepatology* 1989, 9, 380-392], and eventually be
transferred to the gastrointestinal tract and eliminated in feces [Ref. 24. *J. Controlled Release* 2016, 240,
332-348]. The procedure of clearance takes a relative long time to complete, ranging from hours to weeks.
The prolonged hepatobiliary clearance adds to NMs accumulating in the liver for long periods of time,
which increases NMs uptake by liver cells [Ref. 25. *Adv. Mater.* 2022, 34, 2106456].

Furthermore, nanoparticle size is the key factors affecting filtration efficiency, as glomerular filtration
slits in the kidneys are responsible for filtration efficiency. Since the threshold for filtration size of the
glomerular capillary wall is usually 6-8 nm (5.5 nm for metal-based nanoparticles), decreasing the particle
size of inorganic nanoparticles is the first step in improving renal clearance of these particles [Ref. 23.
*Nat. Rev. Mater.* 2018, 3, 358-374; Ref. 27. *ACS Nano* 2015, 9, 6655-6674].

In the biodistribution study mentioned in our previous manuscript (Figure 3.14a), a noticeable
distribution of Gd element in liver was observed. The main reason for this result is that ACNC was
administered with simulated bolus injections (~1 mL/s) to mimic the rapid clinical medication of MR
contrast agent via high-pressure syringe that was administrated in our in vivo studies. ACNC would form
aggregates with larger size due to this injection method, which is difficult to quickly clear from the kidney.
Given this, we performed another pilot evaluation of ACNC biodistribution in vivo with a relatively slow

injection speed ($\sim 500 \mu\text{L/s}$). The result showed that intravenous injection with lower speed could
significantly decrease the accumulation of ACNC in the liver and spleen, which provides valuable guiding
experience for the pre-clinical administration (Figure 3.14b).

**Figure 3.14** The time dependent distribution of ACNC in organs of mice after **a** rapid manual injection or
**b** normal manual injection of ACNC.

In addition to the particle size, surface modification also plays a role in affecting the renal clearance
of engineered nanoparticles. Evidence has shown that nanoparticles with polyethylene glycol (PEG)
modification can significantly lower the adsorption of serum protein and thereby retard the NMs uptake
by RES organs (liver, spleen, etc.) [Ref. 28. *Small* 2011, 7, 271-280; Ref. 29. *Nat. Rev. Drug Discov.* 2003,
2, 214-221; Ref. 30. *Nano Lett.* 2009, 9, 2354-2359]. Notably, zwitterionic glutathione-coated
nanoparticles are excreted through the bladder faster than PEG-coated nanoparticles in the early
elimination phase [Ref. 31. *Angew. Chem. Int. Ed.* 2013, 55, 16039-16043]. Zwitterionic coating
nanoparticles are cleared in a more rapid way than neutral ones, followed by positively charged
nanoparticles [Ref. 32. *Anal. Chem.* 2015, 87, 8941-8948]. Hence, zwitterionic surface ligands
may affect clearance more than single charged surface ligands, even when they share the same net surface
charge [Ref. 33. *Adv. Mater.* 2016, 28, 8162-8168].

There will be cell type-specific responses to the accumulation of NMs in the liver and their uptake
by liver cells. As more effort is put into understanding nano-liver interactions, we could expect to develop
nanoparticles for bioapplications with better safety [Ref. 25. *Adv. Mater.* 2022, 34, 2106456]. To sum up,
optimized injection method and tailored surface modification of nanomaterials can further facilitate their
rapid clearance and help reduce the accumulation in the liver and other organs. There is, of
course, still work to be done to confirm the clearance of ACNC and the interaction with organs and
cells in vivo, and these tests are still ongoing.

References for response letter:

[Ref. 20.] Wilhelm, S., et al. Analysis of nanoparticle delivery to tumours. *Nat. Rev. Mater.* 1, 16014 (2016).

[Ref. 21.] Tsoi, K. M., et al. Mechanism of hard-nanomaterial clearance by the liver. *Nat. Mater.* 15, 1212-
1221 (2016).

[Ref. 22.] Tang, S.H., et al. Tailoring renal clearance and tumor targeting of ultrasmall metal nanoparticles
with particle density. *Angew. Chem. Int. Ed.* 55, 16039-16043 (2016).

[Ref. 23.] Du, B. J., Yu, M. X. & Zheng, J. Transport and interactions of nanoparticles in the kidneys. *Nat.*
*Rev. Mater.* 3, 358-374 (2018).

[Ref. 24.] Zhang, Y.N., Poon, W., Tavares, A.J., McGilvray, I.D. & Chan, W.C.W. Nanoparticle-liver
interactions: Cellular uptake and hepatobiliary elimination. *J. Controlled Release* 240, 332-348 (2016).

[Ref. 25.] Li, J. J., Chen, C. Y. & Xia, T. Understanding Nanomaterial-Liver Interactions to Facilitate the
Development of Safer Nanoapplications, *Adv. Mater.* 34, 2106456 (2022).

- [Ref. 26.] Renaud, G., Hamilton, R.L. & Havel, R.J. Hepatic-metabolism of colloidal gold-low-density
lipoprotein complexes in the rat - evidence for bulk excretion of lysosomal contents into bile. *Hepatology*
**9**, 380-392 (1989).
- [Ref. 27.] Yu, M. X. & Zheng, J. Clearance Pathways and Tumor Targeting of Imaging Nanoparticles. *ACS*
*Nano* **9**, 6655-6674 (2015).
- [Ref. 28.] He, Q. J., Zhang, Z. W., Gao, F., Li, Y. P. & Shi, J. L. In vivo Biodistribution and urinary
excretion of mesoporous silica nanoparticles: effects of particle size and PEGylation. *Small* **7**, 271-280
(2011).
- [Ref. 29.] Harris, J. M. & Chess, R. B. Effect of pegylation on pharmaceuticals. *Nat. Rev. Drug Discov.* **2**,
214-221 (2003).
- [Ref. 30.] Choi, H.S., et al. Tissue- and organ-selective biodistribution of NIR fluorescent quantum dots.
*Nano Lett.* **9**, 2354-2359 (2009).
- [Ref. 31.] Liu, J.B., et al. PEGylation and zwitterionization: pros and cons in the renal clearance and tumor
targeting of near-ir-emitting gold nanoparticles. *Angew. Chem. Int. Ed.* **52**, 12572-12576 (2013).
- [Ref. 32.] Zhu, X.L., et al. Real-time monitoring in vivo behaviors of theranostic nanoparticles by contrast-
enhanced T₁ imaging. *Anal. Chem.* **87**, 8941-8948 (2015).
- [Ref. 33.] Kang, H.M., et al. Renal clearable organic nanocarriers for bioimaging and drug delivery. *Adv.*
*Mater.* **28**, 8162-8168 (2016).

10. In the nanocluster characterisation part (p5), although there is no exact formula for the nanocluster,
the doping percentage of gadolinium in ACNC is still expected to be mentioned for the following analysis

and comparison. How will the amount of added Gd(III) salts affect the overall performance (relaxivity,
toxicity, stability)? There is no discussion on it.

** Thank you for this valuable comment.

As you suggested, the extra study of Gd helped us better evaluate the performance and stability of
ACNC. The amount of substance ratio of Gd and Ca was calculated by measuring the element masses
using ICP-AES method. According to our calculations, the Gd/Ca atomic ratio of ACNC was
approximately 1:3, suggesting the doping percentage of Gd³⁺ in ACNC was to be around 25%.

We further designed and fabricated two more amorphous carbonate nanoclusters with varying ratios
of Gd added for comparison purpose. In the synthesis of ACNC, mol[GdCl₃]: mol[CaCl₂] equals to 1:5.
We also prepared two samples with initial Gd/Ca feed ratio as 1:10 and 1:2 (denoted as ACNC_(Gd/Ca=1:10)
and ACNC_(Gd/Ca=1:2), respectively).

ACNC_(Gd/Ca=1:10) and ACNC_(Gd/Ca=1:2) were both amorphous phases. Moreover, no crystallization
peaks could be found in XRD patterns of the two products dispersed in aqueous solutions over 3 months
(Figure 3.15), indicating that ACNC_(Gd/Ca=1:10) and ACNC_(Gd/Ca=1:2) were stable in aqueous solution over
the long term.

 **Figure 3.15** XRD patterns of fresh ACNC_(Gd/Ca=1:10) and ACNC_(Gd/Ca=1:2), and them staying in aqueous
 solution for 3 months.

 We conducted MTT assay in order to examine the cytotoxicity of these two products. Both HK-2 and
 HaCat cells were incubated with ACNC_(Gd/Ca=1:10) and ACNC_(Gd/Ca=1:2) individually for 24 and 48 hours.
 When the amount of Gd was increased to 500 μg (Gd)/mL, cell viability still remained above 98%,
 indicating good biocompatibilities of both Gd-doped products (Figure 3.16).

 **Figure 3.16** MTT results of **a** ACNC_(Gd/Ca=1:10) and **b** ACNC_(Gd/Ca=1:2) evaluated on HK-2 and HaCaT cells
 at 24h and 48 h.

 The relaxation times of these products were also measured using a 3.0 Tesla MR scanner. The results
 showed that ACNC_(Gd/Ca=1:10) retained a good MR performance with high relaxivity as 34.25 mM⁻¹·s⁻¹
 when the proportion of doping Gd was low. However, when the level of incorporation of Gd was elevated,
 the longitudinal relaxivity (r_1) values of ACNC_(Gd/Ca=1:2) decreased remarkably as 17.21 mM⁻¹·s⁻¹ (Figure
 3.17). One explanation could be that the excessive amount of Gd may potentially perturb the highly
 hydrated ACC-like environment, which in turn affected the generation of ACC with high water content.
 As mentioned in our previous manuscript, the relaxivity of AGC-PAA was as low as 7.91 mM⁻¹·s⁻¹ if only

Gd elements were used to form amorphous nanoclusters. In summary, the doping amount of Gd
significantly affected the relaxivity of paramagnetic ACNC. After comparing two ACNC products with
different Gd doping amounts ($\text{ACNC}_{(\text{Gd}/\text{Ca}=1:10)}$ and $\text{ACNC}_{(\text{Gd}/\text{Ca}=1:2)}$), ACNC reported in our original
manuscript (the initial Gd/Ca feed ratio at 1:5, and a final Gd/Ca atomic ratio at 1:3) was found to be the
best product in terms of contrast performance.

**Figure 3.17** $1/T_1$ versus Gd concentration curve of $\text{ACNC}_{(\text{Gd}/\text{Ca}=1:10)}$ ($R^2 = 0.9993$) and $\text{ACNC}_{(\text{Gd}/\text{Ca}=1:2)}$ (R^2
$= 0.9996$) under 3.0 T.

REVIEWERS' COMMENTS

Reviewer #1 (Remarks to the Author):

The work can be accepted now.

Reviewer #2 (Remarks to the Author):

In this revised manuscript, the authors supplied well-documented evidences to validate the mutual benefit between paramagnetic gadolinium ion and calcium carbonate. The overall performance of final product ACNC and its derived products, including relaxivity, stability and biocompatibility, have been clearly presented. The safety and elimination concerns raised by other referees have also been interpreted by various assessments. The evaluation of the degradation of ACNC under weak acid conditions is imperative and the results are satisfactory. The semiquantitative analysis and relaxivity density further confirmed the excellent imaging performance of this seminal nanoagent. Therefore, I would like to recommend this manuscript to be published in Nature Communications after minor revisions as follows.

1. XPS results of three samples (AGC, AGC-PAA and ACNC) are very important, they should be presented in the main text.
2. The authors should provide the specific values of the time-dependent distribution of ACNC in plasma and the renal clearance efficiency of rats at 24 h.
3. The section in Figure 5a-5c should be provided. Meanwhile, the details should be included in their captions.
4. Please unify the units, please change 'hours' into 'h'; 'minutes' into "min", ...

Reviewer #3 (Remarks to the Author):

The authors have appropriately responded to all questions point-by-point. All unclear claims have been addressed with supportive discussions. Besides, the supplemented results provided more comprehensive studies on the Gd-CaCO₃ nanoclusters and proved their applicability. I would recommend this manuscript be published.

REVIEWERS' COMMENTS

Reviewer #1 (Remarks to the Author):

The work can be accepted now.

**** Thank you so much for recommending acceptance of our work.**

Reviewer #2 (Remarks to the Author):

In this revised manuscript, the authors supplied well-documented evidences to validate the mutual benefit between paramagnetic gadolinium ion and calcium carbonate. The overall performance of final product ACNC and its derived products, including relaxivity, stability and biocompatibility, have been clearly presented. The safety and elimination concerns raised by other referees have also been interpreted by various assessments. The evaluation of the degradation of ACNC under weak acid conditions is imperative and the results are satisfactory. The semi quantitative analysis and relaxivity density further confirmed the excellent imaging performance of this seminal nanoagent. Therefore, I would like to recommend this manuscript to be published in Nature Communications after minor revisions as follows.

1. XPS results of three samples (AGC, AGC-PAA and ACNC) are very important, they should be presented in the main text.

** Thank you for this valuable comment.

We have already added the XPS results and descriptions of three samples (AGC, AGC-PAA and ACNC) in the paper based on your suggestion.

2. The authors should provide the specific values of the time-dependent distribution of ACNC in plasma and the renal clearance efficiency of rats at 24 h.

** Thank you for this valuable comment.

We have already supplied the detailed values of time-dependent distribution of ACNC in plasma (Supplementary Table 6) and the renal clearance efficiency of rats at 24 h (Supplementary Figure 36).

Mouse			Beagle dog		
Time	Mean	SD	Time	Mean	SD
5 min	31.22086	6.60835	5 min	51.48951	4.8424
15 min	15.128	1.8018	30 min	35.77462	2.40858
30 min	10.32743	1.56554	1 h	23.35265	2.79937
1 h	6.64686	1.27997	3 h	12.00936	1.64659
6 h	0.99543	0.18624	6 h	7.65724	1.03522
24 h	0.26514	0.08672	24 h	1.12903	0.28587

Supplementary Table 6. The time dependent distribution of ACNC in plasma of mouse and beagle dog within 24 h (n = 5 biologically independent animals).

Supplementary Figure 36. Renal clearance efficiencies of ACNC in five rats at 24h.

3. The section in Figure 5a-5c should be provided. Meanwhile, the details should be included in their captions.

** Thank you for this valuable comment.

We have already supplied the section information in Figure 5a-5c and their captions.

4. Please unify the units, please change 'hours' into 'h'; 'minutes' into "min", ...

** Thank you for this valuable comment. We have carefully examined the manuscript, and the units were unified.

** In addition, we updated the semiquantitative analysis data in the final version. The semiquantitative analysis was performed in three areas to confirm the enhancement of signal intensity of ACNC groups in comparison to those in Gd-DTPA groups.

Supplementary Figure 29. Semiquantitative analysis. Contrast enhanced MR angiography (MRA) images on a rat and b rabbit, and three statistical areas marked by yellow, blue, red box, respectively. Semiquantitative analysis of the enhancement of signal intensity of (ii) ACNC groups in comparison to those in (i) Gd-DTPA groups on c rat and d rabbit at the immediate (IM) time point from three areas (n = 3 independent areas). The data show means \pm SD.

Supplementary Figure 30. Semi-quantitative analysis. a Sagittal (S) and Coronal (C) MRA images on beagle dog and three statistical areas marked by yellow, blue, red box, respectively. b Semi-quantitative analysis of the enhancement of signal intensity of (ii) ACNC groups in comparison to those in (i) Gd-DTPA groups on beagle dog at the immediate (IM) and 20 s time points from three areas ($n = 3$ independent areas). The data show means \pm SD.

Supplementary Figure 32. Semi-quantitative analysis. MR angiography (MRA) images on a rat and b rabbit and three statistical areas marked by yellow, blue, red box, respectively. Semi-quantitative analysis of the residual signal intensity of ACNC groups on c rat and d rabbit at different time points in comparison to those at 1 min from three areas ($n = 3$ independent areas). The data show means \pm SD.

Supplementary Figure 33. Semiquantitative analysis. MR angiography and rapid renal clearance of ACNC. a MR angiography on beagle dog at different time points after the intravenous injection of ACNC to analyse its dynamic distribution. The kidney continues to brighten within 10 minutes and the bladder (the red arrow) brightened obviously after 10 minutes. b selected areas within the yellow box and c semiquantitative analysis of selected areas to analyse ACNC's dynamic distribution in kidneys and bladder region, respectively (n = 3 independent calculations). The data show means \pm SD. One-way ANOVA with Tukey's multiple comparisons test was used for c to calculate the P value (***) ($P < 0.001$).

Reviewer #3 (Remarks to the Author):

The authors have appropriately responded to all questions point-by-point. All unclear claims have been addressed with supportive discussions. Besides, the supplemented

results provided more comprehensive studies on the Gd-CaCO₃ nanoclusters and proved their applicability. I would recommend this manuscript be published.

** Thank you so much for recommending acceptance of our work.